# Anisotropic carrier dynamics and laser-fabricated luminescent patterns on oriented single-crystal perovskite wafers

Chao Ge [1,2,6] ✉, Yachao Li[1,3,6], Haiying Song [1] ✉, Qiyuan Xie[1], Leilei Zhang[4], Xiaoran Ma[1], Junfeng Liu[5], Xiangjing Guo[1], Yinzhou Yan[1], Danmin Liu[5], Wenkai Zhang [3] ✉, Shibing Liu[1] & Yang Liu [2] ✉

Perovskite materials and their applications in optoelectronics have attracted intensive attentions in recent years. However, in-depth understanding about their anisotropic behavior in ultrafast carrier dynamics is still lacking. Here we explore the ultrafast dynamical evolution of photo-excited carriers and photoluminescence based on differently-oriented $MAPbBr_3$ wafers. The distinct in-plane polarization of carrier relaxation dynamics of the (100), (110) and (111) wafers and their out-of-plane anisotropy in a picosecond time scale were found by femtosecond time- and polarization-resolved transient transmission measurements, indicating the relaxation process dominated by optical/acoustic phonon interaction is related to photoinduced transient structure rearrangements. Femtosecond laser two-photon fabricated patterns exhibit three orders of magnitude enhancement of emission due to the formation of tentacle-like microstructures. Such a ultrafast dynamic study carried on differently-oriented crystal wafers is believed to provide a deep insight about the photophysical process of perovskites and to be helpful for developing polarization-sensitive and ultrafast-response optoelectronic devices.

The research interest in optoelectronics has witnessed an unprecedented surge on organic-inorganic halide perovskites (e.g., $MAPbX_3$, MA = $CH_3NH_3^+$, X = I, Br or Cl), which emerge as leading contenders in solar cells, light-emitting devices, photodetectors, and other applications[1–7]. Albeit most of the optoelectronic applications have been established on polycrystalline thin films, the perovskite single crystals are regarded to represent the upper limits of performance in many of the above-mentioned functional devices[8,9]. From a fundamental viewpoint, the single-crystalline perovskites should reflect the intrinsic physical properties due to their long-range ordered structure and minimal defects, enabling them ideal platforms to scrutinize the inherent attributes of the materials[10,11]. Among all of the intrinsic characteristics of single crystals, anisotropy represents the effect of atomic arrangement symmetry on the directionally dependent properties of a crystalline solid. Anisotropic mechanical, optical, and electronic properties have been well illustrated in various traditional crystalline semiconductors[12–14]; while in terms of halide perovskites, because of the randomly arranged crystalline facets in the polycrystals, it is still in its infancy for the comprehension of their anisotropy especially for the three-dimensional perovskites. Fortunately, in recent years, researchers have investigated the steady-state anisotropy about the structural parameters and physico-chemical properties of

[1]Institute of Laser Engineering, School of Physics and Optoelectronic Engineering, Beijing University of Technology, 100124 Beijing, China. [2]State Key Laboratory of Crystal Materials, Shandong University, 250100 Jinan, China. [3]Department of Physics and Applied Optics Beijing Area Major Laboratory, Center for Advanced Quantum Studies, Beijing Normal University, 100875 Beijing, China. [4]State Key Laboratory of NBC Protection for Civilian, 102205 Beijing, China. [5]Key Laboratory of Advanced Functional Materials, College of Materials Science and Engineering, Beijing University of Technology, 100124 Beijing, China. [6]These authors contributed equally: Chao Ge, Yachao Li. ✉e-mail: gechao@bjut.edu.cn; hysong@bjut.edu.cn; wkzhang@bnu.edu.cn; liuyangicm@sdu.edu.cn

perovskite single crystals, by using angle-resolved Raman spectroscopy, photoconductivity, and photoluminescence measurements to unveil the orientation-dependent performance according to their respective application fields. For example, for MAPbI$_3$ crystal, it has been reported an enhanced photocurrent and ionic conductance on (112) crystal plane compared to that of the (100) plane[15,16]; for CsPbBr$_3$ crystal, its (100) surface was found to demonstrate lower bandgap energy and higher carrier mobility than the (111) surface[17]; our team has also shown the photoresponsive anisotropy to both visible and X-ray radiation with an order of (100) > (110) > (111) on MAPbBr$_3$ crystals[18,19].

To decipher the fundamental photophysics underlying the anisotropy, besides relying on the statistic crystallography structures, probing into the excited state of perovskite crystals by using ultrafast pump–probe techniques should forward dynamic information from a transient perspective. Considering the dynamics of excited carriers can significantly determine both the micro- and macro-physical response of materials in optoelectronic devices, numerous efforts have been donated in this filed[20–23]; however, due to the same reason that the targeted samples were principally polycrystalline thin films or nanocrystals, the transient-state anisotropic response of perovskite single crystals still remains sealed.

From a dynamic view, the excited carriers, excitons, phonons, polarons, and their collective interactions can induce spontaneous ions polarization and transient rearrangement of the lattice structure, thus leading to the possibility of breaking the identified crystal symmetry obtained from the conventional "static" lattice structure[24,25]. Recently, by monitoring the evolution of the wave vector resolved ultrafast electron diffraction intensity, Mohite et al. revealed an ultrafast relaxation of lattice distortion caused by interaction between the electron-hole plasma and the monocrystalline 2D perovskite[26], confirming the light-induced transient structural rearrangement. However, to grasp how the evolution pathways of the excited carriers coordinate with the macroscopic symmetry, access to the ultrafast dynamic process of the excited carriers in different crystallographic orientations of perovskite single crystals would afford the most direct link.

Herein, by virtue of the delicately-grown high-quality MAPbBr$_3$ single crystals and the fabricated differently oriented wafers, we performed polarization-resolved transient transmission experiments and analyzed the anisotropic dynamic evolution related to the transient structural rearrangement under ultrafast relaxation. Subsequently, the femtosecond laser was employed to realize contactless and scalable fabrication of luminescent patterns on the oriented perovskite wafers. The results on the one hand unambiguously demonstrated the in-plane polarization of carrier relaxation dynamics and their out-of-plane anisotropy in a picosecond timescale; and on the other hand provide a mechanism understanding about the photoluminescence enhancement of the femtosecond laser-processed microstructures compared to that of the bulk single crystals with the assistance of X-ray photoelectron spectroscopy (XPS) and transient absorption spectroscopy (TAS). These findings are believed to be significant for the future application of perovskite single crystals, particularly in polarization-sensitive and high-speed optoelectronics.

## Results

### Bulk-sized crystal growth and oriented-wafers fabrication
The bulk-sized MAPbBr$_3$ single crystals were grown in one batch based on our recently invented settled temperature and controlled antisolvent diffusion (STCAD) method[18], as shown in Fig. 1a. The STCAD method is specialized in precise control of heat and mass transport during crystal growth, thus producing high-quality single crystals. The driving force of crystal growth stems from a decrease in solubility attributed to the antisolvent diffusion relying on an adjustable carrier gas flow rate. Meanwhile, the solution supersaturation can be maintained in the metastable zone to restrain spontaneous nucleation and

enable growth on the introduced seeds. The seeds were fixed on a platform connecting to a rotating motor to generate relative motion between the crystal with respect to the solution. The forced rotational motion would bring a thinner boundary layer thickness $\delta_c$ during growth process, which can accelerate the heat transport and the effective stacking of molecules to suppress defect formation and boost crystal quality.

As shown in Fig. 1b, the grown MAPbBr$_3$ single crystals possess high quality, with smooth surfaces and high transparency, and without any inclusions or cracking. This is further confirmed through high-resolution X-ray diffraction (HRXRD) and Laue diffraction measurements (Supplementary Fig. S1), which exhibit symmetrical peak profiles with a small full width at half-maximum (FWHM) and well-defined Laue diffractions. The crystal size is large enough to be processed into differently oriented wafers, i.e., the (100), (110), and (111) wafers, by a diamond wire saw machine (Insets of Fig. 1c). The crystal orientations of the three wafers were further verified by X-ray diffraction (XRD) patterns, wherein the sharp diffraction peaks with uniform Miller indices are in good agreement with the corresponding calculated results (Fig. 1c).

MAPbBr$_3$ is regarded to adopt a cubic $Pm\bar{3}m$ space group at room temperature by assuming the non-centrosymmetric organic cation MA to be randomly oriented in a spherical mode to satisfy the $O_h$ point group on the basis of the X-ray crystallography statistics (Fig. 1d). From a basic structure view, the crystallographic geometric configurations of the (100), (110) and (111) planes exhibit fourfold, twofold, and threefold rotational symmetries at equilibrium, respectively (Fig. 1e and Supplementary Fig. S2). When taking the non-centrosymmetric equilibrium orientation of MA cation into account, the symmetry would be further decreased. From the steady-state perspective, the anisotropy in the electronic and optoelectronic behaviors of the pseudocubic perovskites is already a consensus, although the underlying mechanism is still under debate[27,28]. Whether this anisotropy influences the ultrafast excited-state dynamics has rarely been investigated. Here based on the availability of the high-quality and differently oriented crystal wafers, a transient-state investigation probing into the anisotropic ultrafast relaxation of the photoexcited carrier was conducted aiming to obtain a more in-depth structure–function relations from a dynamic view.

### Anisotropic ultrafast relaxation of photocarrier on oriented wafers
Here, we use the polarization-resolved transient transmission technique to study the anisotropic excited-carrier distributions and dynamical response in MAPbBr$_3$ (Fig. 2a and Supplementary Fig. S3). The following measurements were conducted on oriented wafers fabricated from one bulk-sized crystal to ensure the consistency and reproducibility of the experimental data. A 3.1 eV femtosecond pump laser (above bandgap ~2.3 eV) is used to induce photocarriers, and a polarized probe pulse 1.55 eV arriving at various delay time is adopted to probe the anisotropic dynamical evolution of the photogenerated species. A photoinduced absorption band located around this wavelength in TAS (Supplementary Fig. S4), which emerged under different pump wavelengths (400 nm and 515 nm), is speculated to originate from inter-band transitions of free excitons or localized excitons[29–31]. Given the low amplitude and relatively long lifetime of the photoinduced absorption band, and the "soft" lattice nature of MAPbBr$_3$ crystal, it is most likely that free carriers, excitons, and polarons coexist in the photogenerated states, thus we cannot eliminate the possibilities from other attributions to this band. For detailed assignment analysis, see Supplementary Note 1.

First, the pump fluence-dependent measurement with pump and probe polarization fixed along 0° and 90° was conducted (Fig. 2b). The peak transmission signal, observed at ~1.5 ps ($\Delta T/T_0 \mid_{t=1.5}$), signifies the crucial cooling process of the hot carriers with excess energies, which

is dominated by electron-electron scattering and electron-longitudinal optical (LO) phonon scattering in this stage (Stage I). Figure 2c shows the linear dependence of $\Delta T/T_0 |_{t=1.5}$ on photoexcited carrier densities at a moderate pump fluence (<355 μJ cm⁻² in our experiments). To further explore the carrier decay dynamics, we fit the transient transmittance data at different pump fluence by a double-exponential model, and obtained two stages of the carrier decay process (Fig. 2d and Supplementary Table S1). The first fast decay process is attributed to the rapid decay of the LO phonons to the acoustic phonons (Stage II). Following this process, the hot carriers have cooled almost to the band edge. The second slower decay process is attributed to the interaction and propagation of the low-energy acoustic phonons (Stage III), involving the dynamics of thermalized "cold" carriers near the band edge before their eventual trapping and recombination. Lastly, the cooled carriers will recombine through radiative and/or nonradiative channels. The lifetimes of both decay processes ($\tau_1$ and $\tau_2$) exhibit a nonlinear decrease with the increase of pump fluence, which is due to the enhanced multi-particle coupling caused by the increase of the photoexcited carriers concentration. Under higher fluences (>355 μJ cm⁻²) the carrier decay rate slows down owing to the possible phonon bottleneck effect[32]. Considering the fact that the dynamic and timescale of the photoinduced absorption at 1.55 eV are similar to those of the bleaching band at 575 nm (discussed in Supplementary Note 2 and Supplementary Fig. S11), which reflects the free carriers relaxation, the dynamic process of the excited electrons is illustrated in Fig. 2e. It needs to be pointed out that this fitting is

intended to qualitatively evaluate the relative proportion of the lowest order decay pathways rather than to quantitatively determine the actual dynamics.

The anisotropic carrier relaxations on differently oriented MAPbBr₃ wafers were conducted by the 800 nm probe polarization-dependent measurements with the carrier density $n_0$ of $1.43 \times 10^{15}$ cm⁻³. The low pump excitation density makes the carrier relaxation dynamics be less influenced by extrinsic effects such as the phonon bottleneck effect[32] and the multi-particle Auger-recombination[33]. Figure 2f–h shows the pseudocolor polarization-resolved transient transmission plots of the (100), (110), and (111) wafers. The dynamics of each stage of the relaxation—the first fast cooling stage represented by the peak value ($\Delta T/T_0 |_{t=1.5}$) (Fig. 2i–k), and the two decay stages represented by the lifetimes ($\tau_1$ and $\tau_2$) (Fig. 2l–q and Supplementary Tables S2–4), indeed demonstrated obvious angle-dependence with the evolution of different probe polarizations. Out-of-plane anisotropic excited-state relaxation dynamics appeared on the three oriented wafers. Specifically, in-plane polarization dependence was observed on (100) and (111) wafers, and it evolved in different relaxation stages; interestingly, the (110) wafer showed no polarization-dependent dynamics, being of isotropic at any relaxation stage. We also measured the dynamics evolution on the three wafers at a wavelength of 575 nm that located on the bleaching band in TAS (Supplementary Figs. S4a and S11), to reflect more information about the relaxation processes of free carriers in MAPbBr₃. The results show similar polarization-dependent symmetry, with a milder contrast among different orientations compared to that of the 800 nm detection,

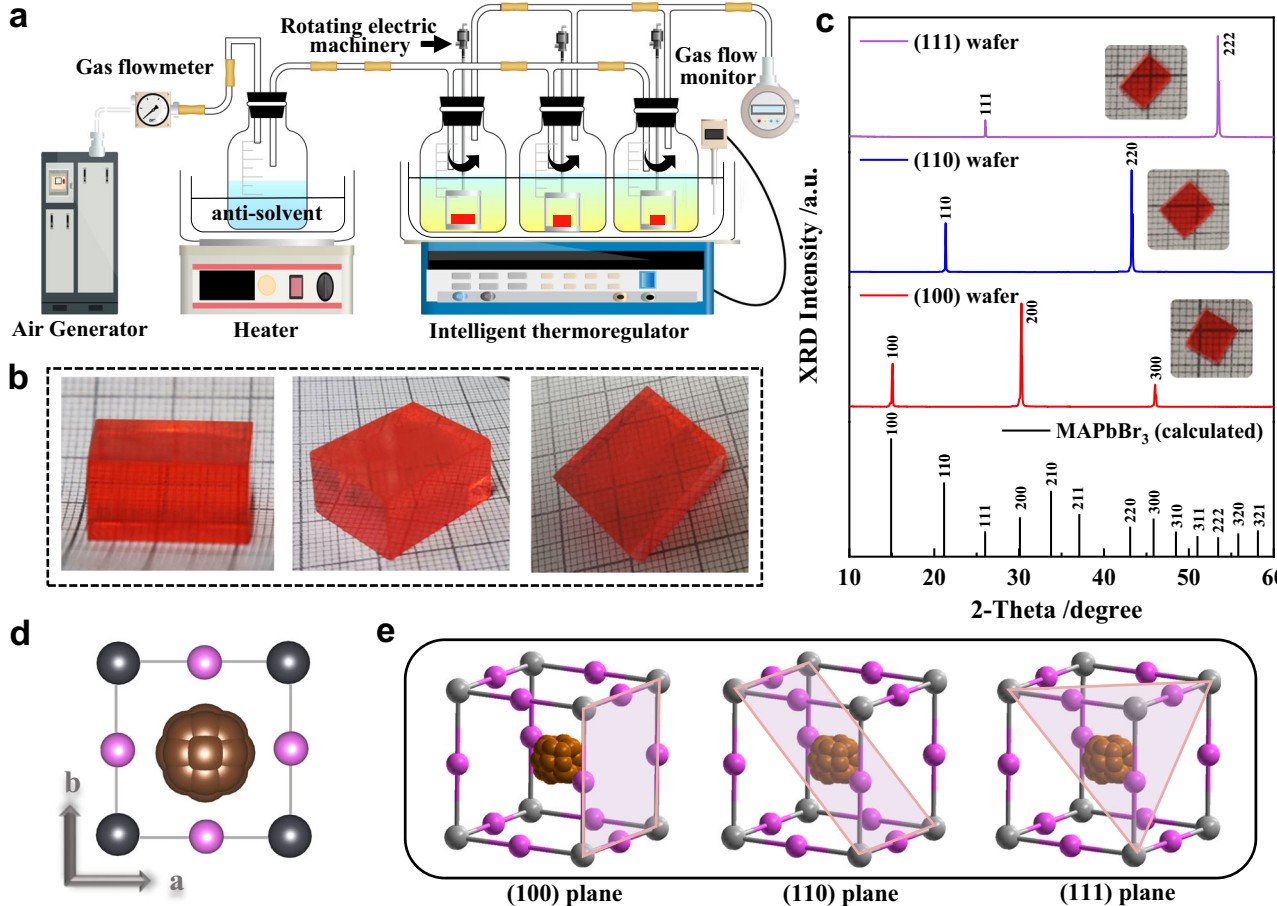

**Fig. 1 | High-quality MAPbBr₃ single-crystal growth and structural characterization of differently oriented wafers. a** Schematic of the STCAD growth apparatus. **b** Photographs of the as-grown MAPbBr₃ single crystals. **c** Measured and calculated X-ray diffraction patterns of differently oriented wafers. The insets show the processed wafers with the same thickness of 1 mm. **d** Crystal structure diagram of MAPbBr₃ viewed along the *c* axis. **e** Crystallographic geometric configurations of the (100), (110), and (111) planes of MAPbBr₃ cubic lattice. (dark gray ball for lead atom, magenta ball for bromine atom, and agminated brown ball for methylamine molecule).

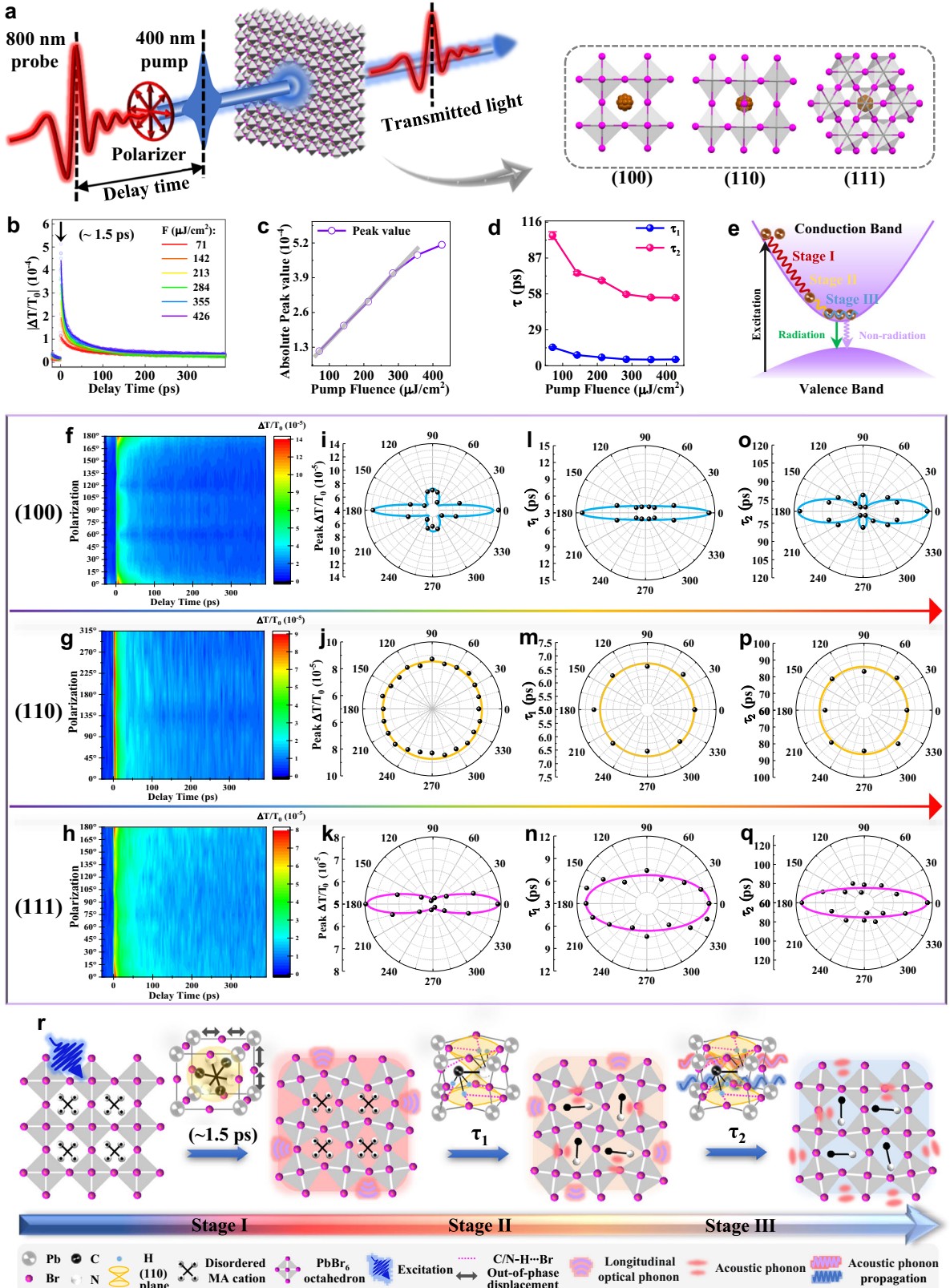

indicating the similar selection rules during the relaxation process regardless of the particle type. For detailed analysis, see Supplementary Note 2.

Based on the above dynamic signatures, the structural changes accompanying the photophysical processes were correlated with the anisotropic carrier relaxation (Fig. 2r). The relaxation can be analyzed as the following three stages according to the involved structural changes. (Stage I) The hot carriers lose their excess energy mainly through polar Fröhlich interactions of electron–LO phonon scattering[34], which is discussed in detail in Supplementary Note 3. It should be noted that the LO phonon mode mainly stems from the out-of-phase stretching of $Pb^{2+}$ and $Br^-$ in the inorganic $PbBr_6$ cage that

**Fig. 2 | Anisotropic relaxation and structure dynamics evolution of MAPbBr₃.**
**a** Schematic illustration of the polarization-resolved transient transmission measurement performed on the differently oriented MAPbBr₃ crystal wafers. **b** The pump fluence-dependent transient transmittance changes $|\Delta T/T_0|$ of the (001) wafer. The results are fitted by the biexponential decay function. The fitting equation is: $|\Delta T/T_0|_t = A_1 e^{(-t/\tau 1)} + A_2 e^{(-t/\tau 2)} + A_0$. $A_1$ and $A_2$ correspond to the respective phonon-mediated relaxation channels; $A_0$ represents the thermal diffusion. **c** The transient transmission peak value $(\Delta T/T_0|_{t=1.5})$ as a function of pump fluence. **d** The decay lifetimes as a function of pump fluence. **e** Schematic of the photoexcited

carrier relaxation dynamics: (Stage I) electron–LO–phonon scattering (Fröhlich interaction); (Stage II) optical phonon decay to acoustic phonons; (Stage III) acoustic phonons decay and propagation. **f**–**h** The pseudocolor polarization-resolved transient transmission plots of the (100), (110) and (111) wafers, respectively. **i**–**k** The probe polarization on the peak value $(\Delta T/T_0|_{t=1.5})$ of differently oriented wafers. **l**–**q** The probe polarization on the decay lifetimes of differently oriented wafers. **r** Schematic of the structural vibrations accompanying the anisotropic carrier relaxation in MAPbBr₃.

governs the carrier relaxation in the first cooling stage. (Stage II) LO phonons transfer the majority of energy to daughter acoustic phonons through energy coupling[35]. The phonon–phonon scattering involved in this process is primarily identified from the co-vibration between organic and inorganic sub-lattices, including the geometric deformation of the octahedra cage and the reorientation of MA which were coupled via H-bonds (C/N-H···Br). (Stage III) The slower relaxation process is attributed to the continued coupling between acoustic phonons and the diffusion to the far-field lattice. During this stage, the distortion of the inorganic octahedra and organic cation began to relax slowly towards the equilibrium structure.

According to the analysis based on the experimental results, we can reasonably speculate that the carrier relaxation process is indeed orientation-dependent in MAPbBr₃. However, due to the incomparable excited-state electronic structure with that of the ground state, the mechanisms behind the ultrafast hot carrier relaxation process are too subtle to be accurately learned. Frankly speaking, we still do not have exact reason about why (110) wafer behaves distinctly in the excited-state carrier relaxations compared to other wafers. Anyhow, we can still find clues from analyzing the distortion of the PbBr₆ octahedral framework and the dynamic orientation of MA⁺ cations, investigating the distribution of defect density along different crystallographic orientations, and taking into account the influence of surface and many-body effects. These factors may affect the excited electronic structure and phonon behavior of MAPbBr₃ single crystal. Thus an azimuthally balanced lattice deformation and a more highly symmetrical distribution of the excited-state charge density on the (110) plane are anticipated to induce polarization-independent excited-state dynamics. See Supplementary Note 4 for detailed analysis. It is important to note that it requires more detailed experimental and theoretical investigations to correlate the observed polarization-dependent dynamics with the crystallographic structures in different crystal planes. Although a comprehensive understanding has not been achieved yet, this discovery holds significant implications as it provides a crystallographic perspective for a deeper understanding of the ultrafast carrier relaxation pathways.

## Laser processing of luminescent micro-patterns on oriented wafers

In view of the anisotropic dynamics of differently oriented MAPbBr₃ wafers induced by femtosecond lasers, we further quantitatively compared the nonlinear absorption (NLA) of the (100), (110), and (111) wafers by employing the well-established transmission open-aperture Z-scan technique (Supplementary Fig. S5). As shown in Fig. 3a, the concave dotted curves fit the two-photon absorption (TPA) theoretical model[36] well, revealing reverse saturable absorption characteristics of MAPbBr₃. According to the Z-scan data, we numerically calculated the effective TPA coefficients as 0.88 cm GW⁻¹ for (110), 0.75 cm GW⁻¹ for (100), and 0.54 cm GW⁻¹ for (111), with an obvious anisotropy of (110) > (100) > (111) on the three differently oriented single-crystal wafers.

Such a strong TPA effect is applied in the femtosecond laser processing to modify the surface structure and subsequently to enhance the optoelectronic properties of MAPbBr₃. Figure 3b illustrates the femtosecond laser two-photon processing of MAPbBr₃

wafers. On the processed region, significantly enhanced photoluminescence (PL) was observed (the laser scanning confocal microscopy (LSCM) images in the insets of Fig. 3b and the fluorescent microscopy (FM) images in Fig. 3e). As shown in Fig. 3c, the femtosecond laser processing endows an enhancement in PL intensity of approximately three orders of magnitude; moreover, the PL enhancement exhibits similar anisotropic characteristic with that of the nonlinear absorption coefficients of the three oriented wafers. Through femtosecond laser processing, scalable luminescent patterns with fine spatial resolution and precision can be realized on the oriented MAPbBr₃ wafers (Fig. 3e and Supplementary Fig. S6). Micro-Raman spectra in Fig. 3d show no obvious peak shifts after processing, indicating that the molecular/lattice structure remains stable. Whereas obvious morphology changes were found in the processed region, wherein many tentacle-like microstructures emerged by the impact of femtosecond laser (Fig. 3f). These microstructures were speculated to respond for the significant PL enhancement[37].

## PL mechanism in bulk crystal and laser-processed microstructure

To verify the determinant role of the microstructures in PL enhancement, we first controlled the presence and distribution density of microstructures by adjusting the processing parameters (e.g., laser power, scanning speed, and defocusing distance) to change the interacting intensity of laser–MAPbBr₃. As shown in Figs. 4a–e, at a processing condition of [laser power: 5 mW, defocus: 470 μm, scanning speed: 2 mm s⁻¹], an obvious groove was generated on the wafer surface with a depth of ~10.3 μm and relative flat processing faces (absence of tentacle-like microstructures). This groove shows no emission under a fluorescence microscope. When the scanning speed was lowered to 1 mm s⁻¹ and 0.1 mm s⁻¹, the femtosecond laser induced deeper grooves (details are seen in Supplementary Fig. S7); and more importantly, tentacle-like microstructures emerged on the rough processing faces. From the PL spectra shown in Fig. 4f, we can see the green emissions arise along with the reduction of the scanning speed. The fluorescence microscopy images (Fig. 4a) more intuitively reflect this boosted emissions in both distribution and intensity to be accompanied by the proliferation of the tentacle-like microstructures induced by enhanced laser effect.

To probe any chemical variations induced by laser processing that may affect the defect types, we conducted X-ray photoelectron spectroscopy (XPS) measurements on both the pristine and laser-processed MAPbBr₃ crystals. Figure 4g, h shows the Pb 4$f$ spectra and the corresponding analytical fitting. The results indicate that there are excess metallic Pb atoms[38] in the pristine MAPbBr₃ (unprocessed crystals), possibly due to the unintended losses of Br atoms or incomplete reaction between MABr and PbBr₂[38,39], although the grown MAPbBr₃ crystals possess a very high crystalline perfection[40]. It has been proved that such metallic Pb atoms would cause deep defect levels in the bandgap, which act as nonradiative decay channels to degrade the photoluminescence[41]. While for the processed MAPbBr₃ crystal, the XPS analysis reveals a dramatic decrease of the metallic Pb peaks; and simultaneously a distinct set of lower binding energy peaks at 138.2 eV and 143.1 eV became remarkable, which can be assigned to Pb coordination with adsorbed

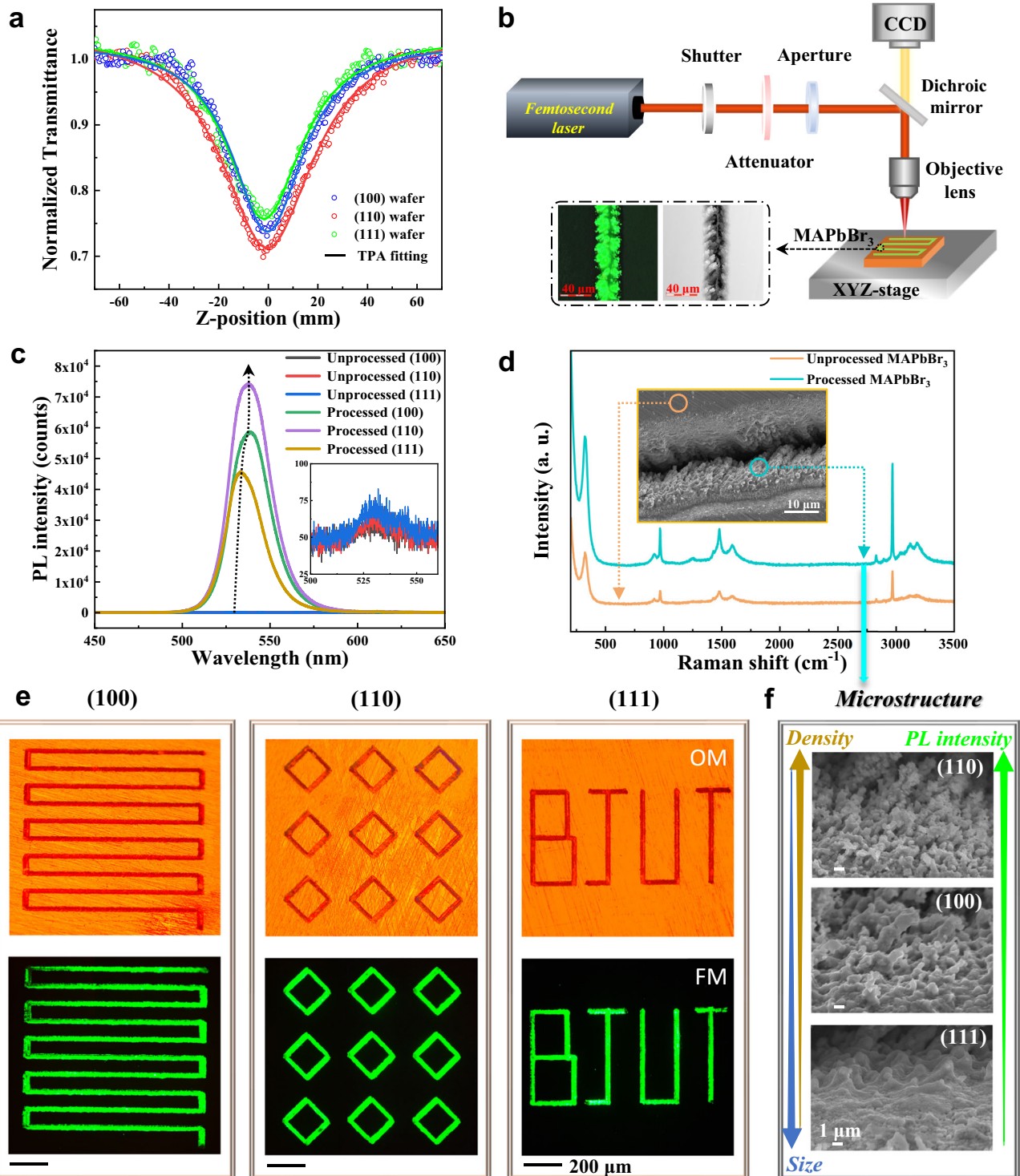

**Fig. 3 | Scalable luminescent patterns and anisotropic enhanced PL on oriented MAPbBr₃ wafers. a** Open-aperture Z-scan measurements performed on the differently oriented MAPbBr₃ single-crystal wafers. The fitting equation of the the theoretical model is: $T_{TPA} = [1 + (n\text{-}1)\,\alpha_n\,L_{eff}\,(I_0/(1 + (z/z_0)^2))^{(n\text{-}1)}]^{-1/(n\text{-}1)}$, $L_{eff} = (1\text{-}e^{-\alpha L})/\alpha$, where $\alpha_n$ represents the effective multiphoton absorption coefficient, $I_0$ is the laser intensity, $z_0$ is the Rayleigh length, $L_{eff}$ is the effective length of the sample, $\alpha$ is the linear absorption coefficient, and $L$ is the actual length of the sample. **b** Schematic illustration of the femtosecond laser two-photon processing measurement performed on oriented MAPbBr₃ crystal. The insets are the fluorescence and height profile images of LSCM. **c** Micro-PL spectra of three MAPbBr₃ wafers before and after processing. **d** Micro-Raman spectra of (001) wafer before and after processing. The inset is the scanning electron microscope (SEM) image of processed MAPbBr₃. **e** The optical microscopy (OM) and FM images of femtosecond laser-processed scalable patterns on three MAPbBr₃ wafers. The processing parameters are [laser power: 5 mW, defocus: 470 μm and scanning speed: 0.1 mm s⁻¹]. **f** The microstructures (SEM) induced by femtosecond laser of the (100), (110), and (111) wafers.

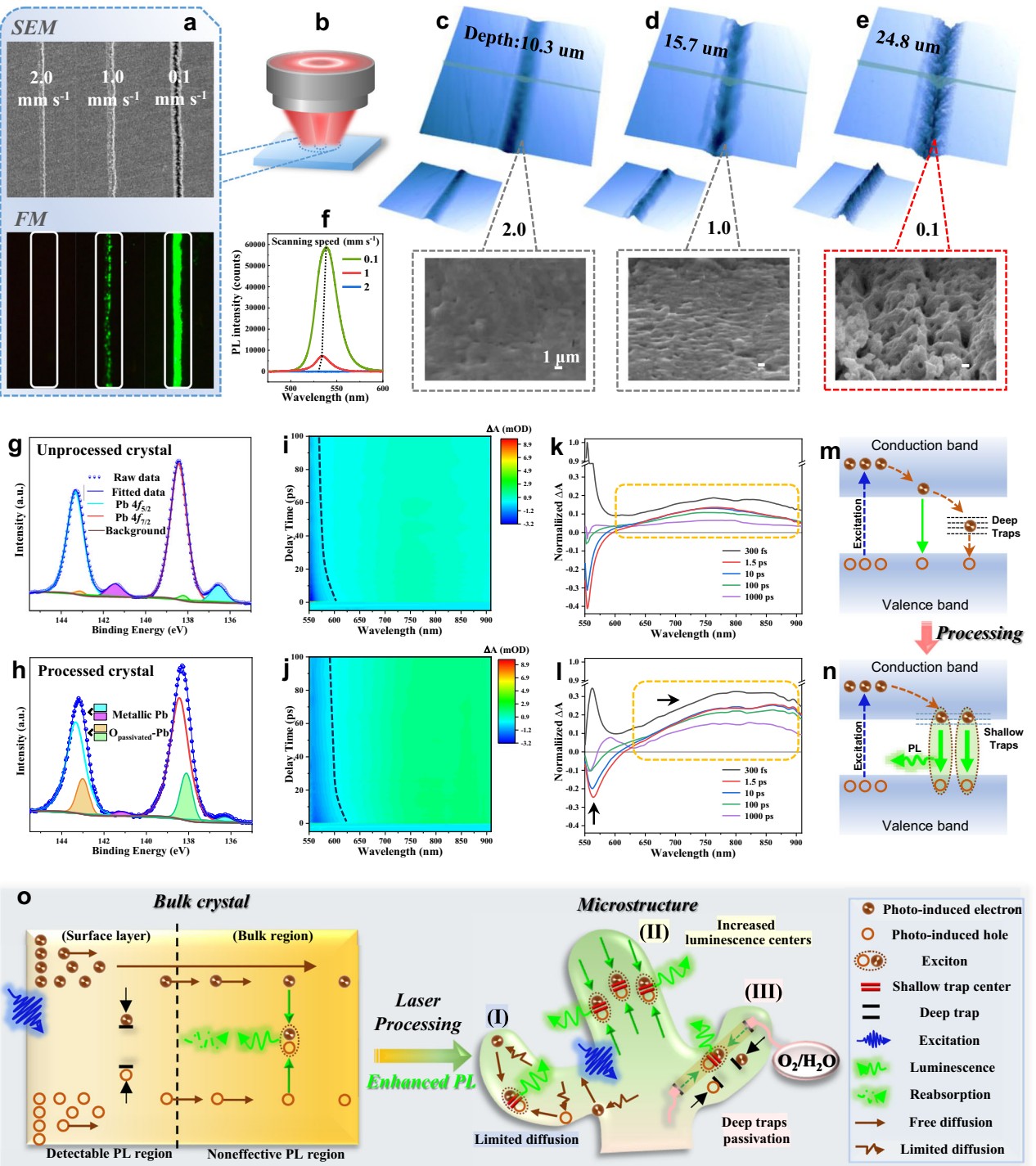

**Fig. 4 | The analysis of PL mechanism in bulk crystal and microstructure of MAPbBr₃.** **a** The SEM and FM images of the processed MAPbBr₃ (001) wafer under different scanning speeds. **b** Schematic diagram of femtosecond laser defocus processing (470 μm before focus). **c–e** The front-view and back-view colored 3D profile images of LSCM and corresponding magnifying SEM images at different processing scanning speeds. **f** Micro-PL spectra of processed MAPbBr₃ at different scanning speeds. **g**, **h** Binding energy spectra and the corresponding analytical fitting for MAPbBr₃ crystals before and after laser processing. **i**, **j** Pseudo color TA plots before and after laser processing with a low excitation carrier density of $1.45 \times 10^{16}$ cm$^{-3}$. **k**, **l** TA spectra before and after laser processing at different delay time. **m**, **n** Flat-band energy level diagram for illustration of different relaxation pathways of the photocarriers in unprocessed (**m**) and processed (**n**) MAPbBr₃ crystals. **o** Schematic diagram of PL driven by structural change before and after processing of MAPbBr₃.

oxygen (denoted as O$_{passivated}$-Pb)[42]. This could be attributed to the effective passivation by ambient air through oxygen binding to the metallic Pb, due to the large specific area surfaces of the micro-/nano-structures formed during laser processing. Accordingly, the non-radiative deep-level traps transformed into radiative shallow traps. The shallow trapped luminescence centers coincide with the

bathochromic shift in emission wavelength compared to that of the pristine bulk crystal as depicted in the PL spectra of Fig. 4f. In addition, the redshifted and enhanced emission of the laser-processed crystals also can be attributed to an increase in the population of band edge excitons owing to the size confinement effect of the microstructure.

To further validate this concept, we performed transient absorption spectroscopy (TAS) on both the pristine and laser-processed MAPbBr$_3$ wafers under identical conditions, as shown in Fig. 4i–l and Supplementary Fig. S9. We used an excitation of 3.1 eV with an excitation density of $1.45 \times 10^{16}$ cm$^{-3}$. It is noted that within a timeframe of 600 fs, a broad absorption band (~550–900 nm) was observed prior to the appearance of the bleach signal (Fig. 4k, l and Supplementary Fig. S8). This photoinduced absorption signal seems to be an inescapably coherent artifact[43], probably due to the two-photon absorption when the pump and probe beams overlap, especially in the bulk crystal. By comparison of the TAS data before and after laser processing, we observed that the intensity of the bleach signal near the bandgap decreased after processing, and the bleach band became broader and redshifted with respect to that before processing, verifying the emergence of shallow trap states energy levels. In addition, the photo-induced absorption band (exciton absorption band) covering a broad long-wavelength spectrum got a significant enhancement by laser processing over 1 ns. These observations indicate a decrease in the population of free carriers and an increase in the population of excitons after laser processing. The presence of shallow trap states energy levels and the redistribution of the population of free carriers and excitons are further verified by intensity-dependent PLQYs (Supplementary Note 5). The simplified flat-band energy level schematic diagram is depicted in Fig. 4m, n.

Based on the observations mentioned above, the significant photoluminescence enhancement after laser processing can be mainly attributed to three factors, the passivation of deep-level traps of metallic Pb atoms by oxygen exposure, the presence of shallow-level traps that act as beneficial recombination centers induced by femtosecond laser processing, and an increase in the population of excitons due to the size confinement effect of the microstructure. Specifically, the last factor refers to that for the laser-processed microstructures, the micron-scale (much smaller than the free carrier diffusion length in single crystal[44]) on the one hand, prominently limits the carrier diffusion within the microstructure to increase the popularity of recombination, and on the other hand, eliminate the reabsorption loss owing to their nanoscale thickness to boost PL intensity; while in the case of the MAPbBr$_3$ bulk crystal, besides the nonradiative deep traps caused by excess metallic Pb atoms, most of the free carriers generated in the surface layer would diffuse inward to the bulk region due to its long carrier diffusion length, rather than recombining into radiative excitons at the surface layer. These factors contribute synergistically to the enhanced emission of the microstructure compared to the pristine bulk crystal, and a more detailed and intuitive model is presented in Fig. 4o.

## Discussion

Based on the differently oriented MAPbBr$_3$ single-crystal wafers, this study presents innovative findings primarily in two aspects. First, utilizing polarization-dependent ultrafast time-resolved spectroscopy, the anisotropic dynamics of photoexcited carriers on the picosecond timescale was first revealed. This discovery provides a deeper understanding of the ultrafast carrier relaxation pathways from the perspective of crystal orientation. It holds significant implications for exploring and expanding the potential of perovskite single crystals in polarization-sensitive and ultrafast optoelectronics applications, such as optical modulators, high-speed light polarization sensors, and ballistic transistors, which require both polarization sensitivity and high-field running capability simultaneously. However, a comprehensive understanding to correlate the observed polarization-dependent dynamics with the crystallographic structures has not been achieved yet due to limitations in current excited-state experimental techniques. Further progress will rely on employing more advanced ultrafast probing techniques, in combination with theoretical simulations, to comprehensively elucidate the observed carrier dynamics behind the underlying excited structure.

Second, by employing femtosecond laser processing, luminescent patterns with a remarkable three-order-of-magnitude PL enhancement on the bulk single crystals were achieved. The observed enhancement can be ultimately attributed to the synergy of three factors: the limited carrier diffusion length, the increase in shallow trap-assisted recombination centers, and the passivation of deep traps within the femtosecond laser-induced tentacle-like microstructures. In addition to offering a convenient top-down strategy for enhancing the photoluminescence intensity of bulk crystals, this study has also provided an in-depth understanding of the luminescence mechanism from multiple spatial (bulk and micro/nanoscale) and temporal (steady and transient-state) dimensions. Considering the delicate relationship between the trap states and the photoluminescence capacity of the 3D perovskite single crystals, looking ahead, further research can delve into the origin of surface defect states to gain a deeper understanding of the luminescence mechanism.

In summary, distinct in-plane polarization of carrier relaxation dynamics and out-of-plane anisotropy in a picosecond timescale were discovered on MAPbBr$_3$ single crystals. The findings are drawn on the basis of high-quality, differently oriented crystal wafers by the transient transmission with polarization-dependent probe measurement. Subsequently, luminescent patterns were fabricated on the wafers by applying femtosecond laser TPA processing. Through TAS and defects inspection, we have elucidated the mechanism behind the three-orders-of-magnitude enhancement in photoluminescence. This study offers a more profound comprehension of the ultrafast carrier dynamics of MAPbBr$_3$, focusing on a crystallographic perspective. We hope such insights will provide more guidance toward the utilization of hot carriers and orientation selection in perovskites optoelectronics.

## Methods

### Growth of MAPbBr$_3$ single crystals

In the growth process, we adopted a precursor (a mixture of MABr and PbBr$_2$ in DMF) concentration of 1.8 mol L$^{-1}$ and an antisolvent of dichloromethane. MAPbBr$_3$ single crystal was grown up by a STCAD method. We fixed the temperature of the growth solution at 38 °C and antisolvent at 32 °C. The introduced seed is fixed on a platform connected to a rotating motor and rotates at a speed of 55 rpm. The crystallizer is sealed with a cup filled with silicone oil. The carrier gas flow was maintained below 2 standard cubic centimeters per minute (sccm). The crystals were grown for about 3 days to ~10 mm, and then the carrier gas flow was adjusted to 6–7 sccm. Subsequently, the airflow can be further increased according to the size of the grown crystal.

### Polarization-resolved transient transmittance measurements

The polarization-resolved transient transmittance apparatus is equipped with a regeneratively amplified Ti:sapphire laser system (Coherent Inc.), which outputs an ultrashort pulse of 35 fs at a repetition rate of 1 kHz with a central wavelength at 800 nm. The strong pump pulse at 400 nm was obtained by second harmonic generation using a nonlinear phase-matched barium borate (BBO) crystal. Polarization-resolved transient transmittance measurement adopts a weak monochromatic laser at 800 nm as probe light. The delay time between pump and probe pulses was realized by moving a motorized time delay line. The spot diameter of the pump beam focused on the sample is ~0.4 mm, which is larger than the ~0.2 mm of the probe beam to ensure that all the probe signals come from the photoexcitation region. The polarization of the probe light is modulated by a 1/4 $\lambda$ wave plate and a polarizer.

## TAS measurements

A Yb:KGW laser (1030 nm, 54 kHz, Light Conversion) is utilized to generate two fundamental light beams. One of these beams is directed to an optical parametric amplifier to produce a high-intensity pump beam at 400 nm. Simultaneously, the other beam is focused on a 5 mm sapphire to generate a low-intensity continuum light serving as the probe beam. A delay stage is employed to control the time delays between the pump and probe beams. The pump and probe beams are spatially overlapped at the target samples, and the transmitted probe light is collected using a charge-coupled device.

## Femtosecond laser two-photon processing

The processing light was born in Ti: sapphire femtosecond laser system (central wavelength: 800 nm, pulse width: 35 fs, repetition frequency: 1 kHz). The laser integral flux is modulated by a neutral density attenuator. The laser beam was focused vertically into the sample surface through an objective lens (Olympus, 10×, NA 0.25) and fixed on a 3D ($x$–$y$–$z$) translation stage (ESP300, Newport Inc.).

## Nonlinear open-aperture $Z$-scan technique

The aforementioned femtosecond laser source is used to measure the nonlinear absorption response. By moving the sample back and forth along the optical axis (defined as the $z$-axis) through an electric translation stage, the laser intensity transmitted through the sample changes to produce differential transmittance. Considering the fitting equations of multiphoton absorption and scattering, the experimental $Z$-scan curve shows the nonlinear absorption characteristics of the sample.

## Micro-Raman and micro-PL measurements

Micro-Raman spectra were measured by a laser confocal Raman spectrometer (Renishaw Inc.) with an excitation wavelength of 785 nm. Micro-PL spectra were measured by a self-built micro-PL spectroscopy. A continuous-wave (CW) laser beam at 409 nm was focused onto the sample surface by a ×10 objective (Olympus, NA 0.25). The spectrum is collected and analyzed by the spectrometer (Horiba iHR550) using a traditional backscattering geometry.

## Data availability

The authors declare that data supporting the findings of this study are available within the paper and the Supplementary Information. Further datasets are available from the corresponding author upon request.

## Code availability

Custom codes used in this article are available from the corresponding author upon request.

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

## Acknowledgements

We are grateful for the support of this work by the National Natural Science Foundation of China (Grants 12304479, 52273185, 51973106, 51972006, U2032112, U2230203, and 2022YFB4600400). C.G. thanks for the support from R&D Program of Beijing Municipal Education Commission (Grant KM202310005005), and the State Key Laboratory of Crystal Materials, Shandong University (Grant KF2208). Y.Liu is thankful for the support from the Distinguished Young Scholars of Shandong Province (ZR2019JQ03), and the Shandong University multidisciplinary research and innovation team of young scholars. C.G. and Y.Liu thank Prof. Hai-ming Zhu, Dr. Guang-liu Ran, and Peng Wang for their help in the TAS analysis.

## Author contributions

C.G. and Y. Li designed and performed most of the experiments. L.Z. conducted high-quality bulk crystal growth. Q.X. and X.G. helped in the femtosecond laser processing experiments. X.M. and Y.Y. participated in the micro-PL measurement. J.L. and D.L. conducted crystal structure analysis. S.L., H.S., and W.Z. helped in dynamic theoretical models analysis. Y. Liu and C.G. conceived and supervised the research. All authors discussed the results and commented on the manuscript.

## Competing interests

The authors declare no competing interests.
