## [Peer Review File · Nature Communications]

REVIEWER COMMENTS

Reviewer #1 (Remarks to the Author):

In this work, authors have explored the ultrafast dynamical evolution of photo excited carriers and photoluminescence (PL) based on MAPbBr₃ high-quality bulk single crystals and differently oriented wafers. The distinct in-plane polarization of carrier relaxation dynamics of the (100), (110) and (111) wafers and their out-of-plane anisotropy in a picosecond time scale were found by femtosecond time- and angle-resolved transient transmission measurements, indicating the relaxation process dominated by optical/acoustic phonon interaction is related to the photoinduced transient structure rearrangements. In this paper, the presentation of figures is well, and organization of manuscript is good. However, more experimental validation and logical discussion is required for supporting the observations. After careful consideration, I have decided to accept your paper with some revisions. I believe that addressing these issues will significantly improve the quality of the paper. I look forward to reviewing the revised manuscript.

1.The introduction lacks a clear and engaging narrative that would draw readers in and help them to understand why the research is important. Please provide a critical review of the relevant literature and ensure that all sources are accurately cited. Please revise the introduction to ensure that the research question is clearly stated and that the significance of the study is explained.

2.Page 6: line 146-149: Authors mentioned that in-plane polarization dependence was observed on (100) and (111) wafers and it evolved in different relaxation stages; interestingly, the (110) wafer showed no polarization-dependent dynamics, being of isotropic at any relaxation stage. Why (110) wafer show no polarization-dependent dynamics?

3.How the crystal structure of MAPbBr₃ is arranged with different orientations? Provide the in-plane and out-of-plane view of MAPbBr₃ crystal structure and correlate it to the ultrafast dynamics.

4.Page 6: line 153-154: The hot carriers lose their excess energy mainly through polar Fröhlich interactions of electron–LO phonon scattering. Fröhlich interaction is a long-range interaction. How is the long-range interaction confirmed here? Will this interaction strength differ along different planes? Same question for the slow relaxation process?

5.Authors have mentioned high quality single crystals. How is the high quality confirmed? Provide the defect density quantification and PL spectra for as grown crystals. Additionally, how the defect density changes in differently oriented wafers.

6.Page 13: line 291-292; The defect-related deep traps would be healed by the trap passivation originating from oxygen/water Diffusion. How is the oxygen/water diffusion confirmed? Provide the XPS spectra before and after laser processing?

7.Additionally, please address any limitations of the study and suggest directions for future research and highlight the significance of the findings.

Reviewer #2 (Remarks to the Author):

Ge et al report on two independent observations on oriented single crystals of methylammonium lead bromide: one on the anisotropy of transient dynamics and on the effect of laser processing on the photoluminescence efficiency. While the presented observations are interesting, I have several concerns with the presented conclusions and thus cannot recommend publication in current form.

1. It is critical to note that the single crystals are semiconductors with well defined band structure and the representation in Figure 2e is incorrect. The photoexcitation creates a population of electrons (holes) in the conduction (valence) band, which thermalize to the bottom of the band in a few hundreds of femtoseconds through phonon scattering. The longer time dynamics are a results of recombination, through both radiative and non-radiative (primarily defects) channels. The observed anisotropy has to be discussed within this well established photophysical scenario. The absence of anisotropy in the (110) crystal must be properly discussed.

2. A very important aspect is what one is probing at 1.55 eV? From the TA spectra in figure 4, it appears that the authors are probing bleach dynamics of mid-gap states. How can anisotropy be explained for spectral features of defect states, that are localized and randomly spread over the bulk of the crystal? To be more precise, one must do the experiment at the bandedge.

3. What is effect of the femtosecond laser in the second part of the manuscript? It appears that the top part of the crystal is getting ablated by the laser (through two photon absorption as the authors point out). It is known that the surface of bulk perovskite crystals are intrinsically defective and by ablation, one is revealing the lesser defective bulk phase of the crystal. See Applied Physics Reviews, 6, 031401.

4. The TA analysis presented in figure 4 has a few issues.

a. Firstly, if there is chirping (which looks negative?), it must be properly corrected. Moreover, I don't think the bleach is appearing later than the sub-gap states. The authors are creating a population way above in the CB and thermalization results in a few picoseconds in the appearance of the GSB at the bandedge (see Nature Photonics, 9, 695).

b. The narrow spectral feature at around 703 nm in Figure 4h appears to be an artefact, most likely from the saturation of the detector. The authors should show the transmission spectrum of the white light taken at the same condition as the measurement to discount this effect. T

c. here is no convincing data to substantiate the model in figure 4j and n. Intensity dependent dynamics should help here. However, looking at the TA intensity, the authors are likely to be in the high density Auger regime and the dynamics is not going to representative of the defect dynamics.

Response to the Comments and Suggestions of Reviewer 1

The reviewer's comments: *In this work, authors have explored the ultrafast dynamical evolution of photo excited carriers and photoluminescence (PL) based on MAPbBr₃ high-quality bulk single crystals and differently oriented wafers. The distinct in-plane polarization of carrier relaxation dynamics of the (100), (110) and (111) wafers and their out-of-plane anisotropy in a picosecond time scale were found by femtosecond time- and angle-resolved transient transmission measurements, indicating the relaxation process dominated by optical/acoustic phonon interaction is related to the photoinduced transient structure rearrangements. In this paper, the presentation of figures is well, and organization of manuscript is good. However, more experimental validation and logical discussion is required for supporting the observations. After careful consideration, I have decided to accept your paper with some revisions. I believe that addressing these issues will significantly improve the quality of the paper. I look forward to reviewing the revised manuscript.*

Authors reply: We sincerely thank the reviewer for the positive assessment and summary of our work, and giving us constructive suggestions to improve the quality of the manuscript.

1. *The introduction lacks a clear and engaging narrative that would draw readers in and help them to understand why the research is important. Please provide a critical review of the relevant literature and ensure that all sources are accurately cited. Please revise the introduction to ensure that the research question is clearly stated and that the significance of the study is explained.*

Authors reply: Thanks for the reviewer's constructive comment. In response to the reviewer's suggestion, we have extensively revised the introduction to provide a more comprehensive research background and emphasize the significance of our study. **The revised content has been incorporated into the Introduction Section of the main text and is highlighted in red for easy identification.**

2. *Page 6: line 146-149: Authors mentioned that in-plane polarization dependence was observed on (100) and (111) wafers and it evolved in different relaxation stages; interestingly, the (110) wafer showed no polarization-dependent dynamics, being of isotropic at any relaxation stage. Why (110) wafer show no polarization-dependent dynamics?*

Authors reply: We sincerely thank the reviewer's constructive suggestion. Indeed the in-plane polarization-dependent dynamics on the (100) and (111) wafers and the isotropic dynamics on the (110) wafer represent one of the most prominent discovery revealed by this study. In the current experiment, the relaxation time scale mainly concerns the carrier dynamics involving hot carrier cooling, polaron formation, exciton formation, and the

subsequent dynamics of “cold” carriers near the band-edge involving photoinduced lattice expansion, strain, and coherent phonon effects. During these photophysical processes, the carrier–phonon interaction plays a significant role by inducing local lattice displacements which generate a polarization-induced electric field that in turn interacts with the charge carriers. Thus the involvement of phonons (lattice vibrations) in all these stages correlates the carrier relaxation dynamics with the lattice structure.

From a structural view, although MAPbBr₃ is regarded as a highly symmetric cubic structure at room temperature resting on identification of the organic MA⁺ cation as having random orientations in the X-ray crystallography statistics,¹ our group and other researchers have indeed revealed in-plane and out-of-plane structure anisotropy through angle-resolved Raman spectroscopy, as well as orientation-dependent optoelectronic properties by using photoconductivity and photoluminescence measurements.²⁻⁵ These recent investigations unambiguously corroborated the steady-state anisotropy on MAPbBr₃ single crystals, however, the transient-state anisotropy in the ultrafast carrier dynamics still remains sealed. This is because the anisotropy of steady-state properties relies directly on the static crystal structure that we can depict intuitively; while for the ultrafast carrier dynamics occurring in the excited state, there are two main obstacles. Firstly, photoexcitation can lead to a distinct excited electronic structure compared to the ground state (so far, there is still a big challenge to accurately capture the excited electronic structures experimentally). Secondly, the carrier relaxation process is heavily influenced by phonons, which are hard to describe with a clear picture. Therefore, it is infeasible to directly correlate the carrier relaxation dynamics with the electronic structure of the ground state. In spite of these challenges, we can still find clues from analyzing the distortion of the PbBr₆ octahedral framework and the dynamic orientation of MA⁺ cations. Additionally, investigating the defect density distribution along different crystallographic orientations and considering the surface and many-body effects can provide further understanding. These factors may affect the excited electronic structure and phonon behavior of MAPbBr₃ single crystal.

Lattice deformation, vibration and relaxation effect: Because of the different crystallographic geometric configurations of the (100), (110) and (111) planes (**Figure R1**, including the atomic density, interplanar spacing, etc.) and the distribution of MA⁺ orientation domains, the equilibrium out-of-plane and in-plane anisotropic structure should induce non-synchronized lattice deformation and relaxation upon photoexcitation, which may break the original lattice symmetry in the excited state structure. E.g., the MA⁺ rocking and twisting vibration modes demonstrate distinct in-plane anisotropy on (100) and (110) wafers, which may in turn cause interactions on the distortion of the PbBr₆⁴⁻ inorganic skeletons; and the transient photoluminescence lifetimes also show discrepancies among (100), (110) and (111) wafers.² Consequently the incoherent lattice vibration amplitude and frequency dictate the specific phonon modes along different crystallographic orientations

in MAPbBr₃. Thus an azimuthally balanced lattice deformation on the (110) plane is imagined to induce a polarization-independent excited state dynamics on the (110) crystal plane.

Trap density distribution and surface effect: Because on the different crystallographic planes, the defect formation energies of vacancies and interstitials are different; and likewise, the ion migration also depends on the crystallographic orientations, these will lead trap density distributions inhomogeneously on different crystal wafers. In **Figure R7 & R8**, we have demonstrated the discrepancy of trap densities on differently oriented wafers. Carrier scattering and trapping resulting from defects play a crucial role in influencing carrier relaxation in different orientations within the bulk crystal. Additionally, the polaronic nature of the relatively soft lattice in MAPbBr₃ introduces surface dipoles that affect electronic structure and charge distribution. These factors also become significant determinants of carrier transport and recombination properties on distinct crystal planes. Thus a more highly symmetrical distribution of the excited-state charge density on the (110) crystal plane is anticipated.

Many-body effect: The interactions among electrons and with other particles, such as phonons (lattice vibrations) and other carriers, can lead to many-body effects. These effects cause a redistribution of momentum and energy of the excited state electrons through energy exchange or resonance coupling between the particles. Many-body effect has significant consequences on the carrier's lifetime and optical properties such as absorption and refraction in the excited state, thus making the carrier dynamics even more complicated.

Therefore, due to the incomparable excited state electronic structure with that of the ground state, the mechanisms behind ultrafast hot carrier relaxation process are too subtle to be accurately learned. Frankly speaking, we still do not have exact answers about why MAPbBr₃ (110) wafer behaves distinctly in the excited-state carrier relaxations compared to other wafers. It is important to note that it requires more detailed experimental investigation and deeper analysis to correlate the observed polarization-dependent dynamics with the crystallographic structures in different crystal planes. This study, based on the differently-oriented MAPbBr₃ single-crystal wafers, firstly penetrated into the orientation-dependent dynamical evolution of the excited carriers and provided solid observations of the anisotropy in the ultrafast carrier dynamics. Although a comprehensive understanding has not been achieved yet due to limitations in excited-state experimental techniques, this discovery holds significant implications as it provides a novel perspective for a deeper understanding of the ultrafast carrier relaxation pathways and opens up new possibilities for utilizing perovskite single crystals in polarization-sensitive photoelectron responses.

The above discussions are added to the main text and supporting information file, and are highlighted in red.

References

- 1 Nandi P, Pandey S K, Giri C, et al. Probing the electronic structure of hybrid perovskites in the orientationally disordered cubic phase [J]. *The Journal of Physical Chemistry Letters*, **2020**, *11*(14): 5719-5727.
- 2 Zhang L, Cui S, Guo Q, et al. Anisotropic Performance of High-Quality MAPbBr₃ Single-Crystal Wafers [J]. *ACS Applied Materials & Interfaces*, **2020**, *12*(46): 51616-51627.
- 3 Zhang L, Liu Y, Ye X, et al. Exploring anisotropy on oriented wafers of MAPbBr₃ crystals grown by controlled antisolvent diffusion [J]. *Crystal Growth & Design*, **2018**, *18*(11): 6652-6660.
- 4 Zuo Z, Ding J, Zhao Y, et al. Enhanced optoelectronic performance on the (110) lattice plane of an MAPbBr₃ single crystal [J]. *The Journal of Physical Chemistry Letters*, **2017**, *8*(3): 684-689.
- 5 Yang H, Zhou Y, Yang Y, et al. Crystal facet engineering induced anisotropic transport of charge carriers in a perovskite [J]. *Journal of Materials Chemistry C*, **2018**, *6*(43): 11707-11713.

3. *How the crystal structure of MAPbBr₃ is arranged with different orientations? Provide the in-plane and out-of-plane view of MAPbBr₃ crystal structure and correlate it to the ultrafast dynamics.*

Authors reply: We sincerely thank the reviewer's constructive suggestion. At room temperature, MAPbBr₃ adopts a cubic $Pm\bar{3}m$ space group based on X-ray crystallography statistics. The crystal structure of MAPbBr₃ consists of an inorganic framework of corner-sharing PbBr₆ octahedra and organic MA⁺ cations. The MA⁺ cations are located in the interstitial spaces between the PbBr₆ octahedra, which are assumed to be randomly oriented in a spherical manner to satisfy the O_h point group. The arrangement of the anions and cations, the atom density and the interplanar spacing can vary depending on the crystal orientation. Here is a brief description of the crystal structure for different orientations in **Figure R1**:

(100) wafer: In this wafer, the crystal planes are vertical to the [100] orientation. The crystallographic geometric configuration of the (100) plane exhibits 4-fold rotational symmetry.

(110) wafer: In this wafer, the crystal planes are vertical to the [110] orientation. The crystallographic geometric configuration of the (110) plane exhibits 2-fold rotational symmetry.

(111) wafer: In this wafer, the crystal planes are vertical to the [111] orientation. Compared to the [100] and [110] orientations in MAPbBr₃, the [111] orientation achieves the highest packing density. The crystallographic geometric configuration of the (111) plane exhibits 3-fold rotational symmetry.

Figure R1. The in-plane and out-of-plane views of MAPbBr₃ crystal structure with different orientations. The in-plane (top) views of the MAPbBr₃ crystal structure reveal two-dimensional arrangements forming a continuous network. The out-of-plane (side) views of the MAPbBr₃ crystal structure display the stacking of (100), (110), and (111) planes, respectively, resulting in three-dimensional arrangements.

Indeed the structure of a material has significant impacts on carrier dynamics, through the ways such as charge transport and phonons. For example, a highly ordered and well-defined structure can facilitate charge transport, while disorders or structural defects can impede charge transport. Phonons are quantized vibrations or wave-like motions of atoms in a crystal lattice, whose types, distributions and lifetimes are determined by the specific atomic arrangements and bonding characteristics in the crystal structure, and have important effects on carrier scattering and cooling. However, besides the crystal structure, the ultrafast carrier relaxation dynamics are also influenced by many other factors, as the answers to the above question.

For MAPbBr₃ crystals, the (100), (110), and (111) planes exhibit in-plane rotational symmetries of 4-fold, 2-fold, and 3-fold, respectively, resting on identification of the organic MA⁺ cation as random orientations in the X-ray crystallography statistics. The crystal plane symmetry does not coincide with the polarization-dependent carrier relaxation dynamics. If considering the MA⁺ rocking and twisting vibration which may in turn cause interactions on the distortion of the PbBr₆⁴⁻ inorganic skeletons, the crystallographic symmetry would be decreased. Furthermore, the ultrafast carrier

relaxation dynamics are influenced by various factors including the lattice vibration (phonon- and polaron-related effects) and defect traps, which are all related to the crystallography structure but meanwhile are too complicated to decipher. Thus so far, a clear mechanism to correlate the crystal structure to the ultrafast carrier relaxation dynamics has not been established. This study has taken a first step by revealing the anisotropic carrier dynamics based on the oriented single-crystal perovskite wafers. In the following works we will replenish thoroughly ultrafast probing techniques with the aid of theoretical simulations to fully understand and explain the observed carrier dynamics.

The above discussions are added in the supporting information file, and Figure R1 is added as Figure S2.

4. *Page 6: line 153-154: The hot carriers lose their excess energy mainly through polar Fröhlich interactions of electron-LO phonon scattering. Fröhlich interaction is a long-range interaction. How is the long-range interaction confirmed here? Will this interaction strength differ along different planes? Same question for the slow relaxation process?*

Authors reply: Thanks for the constructive suggestion. As we know, the Fröhlich model addresses the electrons in ionic crystals or polar semiconductors.¹ The strength of the Fröhlich interaction in a material is directly linked to the polar nature of its crystal lattice. In a highly polar material the Coulomb field of a carrier (or exciton) couples more easily to the polar vibrations (*i.e.* LO phonons) of the lattice, resulting in strong Fröhlich coupling. In the structure of MAPbBr₃ crystal, it consists of two interpenetrating sublattices: an inorganic lattice composed of corner-shared PbBr₆⁴⁻ octahedral and a second sublattice composed of MA⁺ cations. The non-superimposed arrangement of the positive and negative charges leads to the existence of electric dipoles in the lattice and makes the perovskite lattice polar. Due to the polar nature of MAPbBr₃, Fröhlich interaction is the dominant relaxation pathway for hot carriers, wherein the electron-longitudinal optical (LO) phonon scattering arises from Coulomb interactions between the electrons and the macroscopic electric field induced by LO phonon mode. Thus the Fröhlich interaction that governs electron-lattice coupling is considered to be long-range interaction. And it has been widely accepted that the Fröhlich interaction is crucial in describing carrier behavior in the polar lattice of lead halide perovskites.²⁻⁵

The electron-phonon interactions lead to the formation of a polaron state, where an electron or a hole deforms the lattice in its vicinity and becomes more localized.⁶ To date,

the conventional Fröhlich interaction has been mainly considered to be the polaron formation mechanism in halide perovskites. Depending on the range and strength of the electron–phonon interaction, polarons can be generally categorized into large and small polarons. For the 3D lead halide perovskites, large polarons are formed by the long-range electron-LO phonon interaction, *i.e.*, the charge carriers coupling to the vibrational motion of the inorganic lattice, while the A site cation only plays a minor and indirect role by affecting the distortion of the inorganic lattice. (Conversely, a small polaron is formed by strong and short-range electron–phonon interactions and is usually localized within a single lattice constant. Small polarons are mostly revealed in the low-dimensional halide perovskites and double perovskites. The Fröhlich interaction model is inapplicable to the strong electron–phonon coupling of small polarons.)

The long-range nature of Fröhlich interaction has been confirmed through several experimental and theoretical evidences.⁷⁻⁹ In particular, Zhu et al.¹⁰ have provided a direct time domain view of large polaron formation in single-crystal MAPbBr₃ and CsPbBr₃ using time-resolved optical Kerr effect (TR-OKE) spectroscopy and in conjunction with hybrid density functional theory (DFT) calculations, obtaining the electron and hole polaron mobilities of $\mu_e = 149.8 \text{ cm}^2 \text{ V}^{-1} \text{ s}^{-1}$ and $\mu_h = 79.2 \text{ cm}^2 \text{ V}^{-1} \text{ s}^{-1}$ with corresponding polaron radii of $\rho_e = 4.18 \text{ nm}$ and $\rho_h = 3.13 \text{ nm}$ in MAPbBr₃. (**Figure R2a**) Lindenberg et al.¹¹ visualized excitation-induced strain fields in MAPbBr₃ via femtosecond resolution diffuse X-ray scattering measurements, confirming the formation of large polarons with a polaron radius of $\sim 3 \text{ nm}$ at $t = 20 \text{ ps}$. (**Figure R2b**) Consequently, the observation of large polarons, which extend over more than a few lattice sites, provides further evidence supporting the existence of long-range Fröhlich interaction in MAPbBr₃ crystals.

Figure R2. Large polarons in MAPbBr₃ crystals. (a) Time-resolved optical Kerr effect (TR-OKE) spectroscopy and hybrid density functional theory (DFT) calculations on MAPbBr₃.¹⁰ (b) Femtosecond resolution diffuse X-ray scattering measurements and schematic of the three-dimensional polaron model on MAPbBr₃.¹¹

About whether the interaction strength will differ along different planes, because the electron–phonon scattering relates to carrier effective mass, LO phonon energy, and dielectric constant, we could find the dependence between crystallographic planes and the strength of electron-phonon coupling interaction according to the so-called Fröhlich constant α is defined by,¹²

$$\alpha = \frac{e^2}{\hbar\omega_0} \left(\frac{1}{\varepsilon_\infty} - \frac{1}{\varepsilon_s} \right) \sqrt{\frac{m\omega_0}{2\hbar}}$$

Where ε_∞ and ε_s are optical and static dielectric constants, respectively, m is the effective mass of the electron in the case of no electron-phonon interaction (bare band mass), and ω_0 describes the LO phonon frequency.

The quantity $\frac{1}{\varepsilon} = \left(\frac{1}{\varepsilon_\infty} - \frac{1}{\varepsilon_s} \right)$, which is known as the dielectric contrast, quantifies the ionic nature of a material and thereby determines the strength of the carrier–lattice interaction. Consequently, the static and high-frequency values of the dielectric function provide a means of evaluating the Fröhlich interaction among differently oriented planes.

Thus we analyzed the dielectric function of differently oriented MAPbBr₃ wafers by measurements of the refractive index (k) and the extinction coefficient (n) through a spectroscopic ellipsometry (SE) method. (Figure R3a & b) The complex refractive index, $N = n - ik$, contains the same information as the dielectric function, $\varepsilon = \varepsilon_1 - i\varepsilon_2$, with $\varepsilon_1 = n^2 - k^2$ and $\varepsilon_2 = 2nk$.¹³ By analysis of n and k as a function of wavelength, it is notable that for both real (ε_1) and imaginary (ε_2) dielectric function in the optical region, the values indeed show discrepancies for differently oriented wafers. (Figure R3c & d) Deducing from the dielectric mechanism, the dielectric contrast should be even larger in the low-frequency THz region where the LO phonons typically reside. Therefore, on account of the measured $\varepsilon_{1(110)} > \varepsilon_{1(100)} > \varepsilon_{1(111)}$, the difference of dielectric function on different crystal planes will definitely affect the Fröhlich interaction strength along different planes.

Figure R3. Dielectric function anisotropy of differently oriented MAPbBr₃ single-crystal wafers. (a) Refractive index (n), **(b)** extinction coefficient (k), **(c)** real (ϵ_1) and **(d)** imaginary (ϵ_2) part of the dielectric function on (100), (110), and (111) oriented wafers.

According to the experimental analysis in this work and the literature reports,¹⁴ there exist three distinct relaxation regimes in the typical lead halide perovskites: (1) carrier-phonon scattering (Fröhlich interaction); (2) optical phonon decay to acoustic phonons; (3) acoustic phonon propagation to the far-field region in the material. The Fröhlich interaction occurs predominantly between the hot carriers and inorganic sub-lattice in the first fast cooling stage, exciting the high-energy lead-halide LO phonons¹², where the long-range interaction can quickly help the hot carrier to dissipate their excess energy. However, the later slower relaxation process is dominated by acoustic phonon scattering, as determined mainly by the short-range deformation potential.¹⁵ Due to the soft lattice nature of halide perovskites, following the fast Fröhlich interaction process, the subsequent evolution of the phonon mode still induce structural modifications, such as lattice expansion and vibration, which are heavily dependent on the crystallographic lattice structure. Thus it is reasonable to speculate that both the long-range Fröhlich interaction of the hot carriers coupling to the inorganic sub-lattice occurring in the first fast cooling stage, and the subsequent slow relaxation process, are related to the lattice arrangement on different crystallographic planes. This is coincident with the crystal plane-dependent anisotropic carrier dynamics revealed in this work.

The above discussions are added in the supporting information file, and Figure R3 is added as Figure S11.

References

- 1 Yamada Y, Kanemitsu Y. Electron-phonon interactions in halide perovskites [J]. *NPG Asia Materials*, **2022**, *14*(1): 48.
- 2 Frost J M, Whalley L D, Walsh A. Slow cooling of hot polarons in halide perovskite solar cells [J]. *ACS energy letters*, **2017**, *2*(12): 2647-2652.
- 3 Iaru C M, Geuchies J J, Koenraad P M, et al. Strong carrier–phonon coupling in lead halide perovskite nanocrystals [J]. *ACS nano*, **2017**, *11*(11): 11024-11030.
- 4 Zhu X Y, Podzorov V. Charge carriers in hybrid organic–inorganic lead halide perovskites might be protected as large polarons [J]. *The Journal of Physical Chemistry Letters*, **2015**, *6*(23): 4758-4761.
- 5 Li M, Bhaumik S, Goh T W, et al. Slow cooling and highly efficient extraction of hot carriers in colloidal perovskite nanocrystals [J]. *Nature communications*, **2017**, *8*(1): 14350.
- 6 Zheng F, Wang L. Large polaron formation and its effect on electron transport in hybrid perovskites [J]. *Energy & Environmental Science*, **2019**, *12*(4): 1219-1230.
- 7 Quarti C, Grancini G, Mosconi E, et al. The Raman spectrum of the CH₃NH₃PbI₃ hybrid perovskite: interplay of theory and experiment [J]. *The journal of physical chemistry letters*, **2013**, *5*(2): 279-284.
- 8 La-o-Vorakiat C, Xia H, Kadro J, et al. Phonon mode transformation across the orthorhombic–tetragonal phase transition in a lead iodide perovskite CH₃NH₃PbI₃: a terahertz time-domain spectroscopy approach [J]. *The journal of physical chemistry letters*, **2016**, *7*(1): 1-6.
- 9 Leguy A M A, Goñi A R, Frost J M, et al. Dynamic disorder, phonon lifetimes, and the assignment of modes to the vibrational spectra of methylammonium lead halide perovskites [J]. *Physical Chemistry Chemical Physics*, **2016**, *18*(39): 27051-27066.
- 10 Miyata K, Meggiolaro D, Trinh M T, et al. Large polarons in lead halide perovskites [J]. *Science advances*, **2017**, *3*(8): e1701217.
- 11 Guzelturk B, Winkler T, Van de Goor T W J, et al. Visualization of dynamic polaronic strain fields in hybrid lead halide perovskites [J]. *Nature materials*, **2021**, *20*(5): 618-623.
- 12 Devreese J T. Polarons[M]//Encyclopedia of applied physics: vol. 14. **1996**: 383-409.
- 13 Löper P, Stuckelberger M, Niesen B, et al. Complex refractive index spectra of CH₃NH₃PbI₃ perovskite thin films determined by spectroscopic ellipsometry and spectrophotometry [J]. *The journal of physical chemistry letters*, **2015**, *6*(1): 66-71.
- 14 Yang J, Wen X, Xia H, et al. Acoustic-optical phonon up-conversion and hot-phonon bottleneck in lead-halide perovskites [J]. *Nature communications*, **2017**, *8*(1): 14120.
- 15 Brennan K F. The physics of semiconductors: with applications to optoelectronic devices [M]. Cambridge university press, **1999**.

5. *Authors have mentioned high quality single crystals. How is the high quality confirmed? Provide the defect density quantification and PL spectra for as grown crystals. Additionally, how the defect density changes in differently oriented wafers.*

Authors reply: According to the reviewer's suggestion, in order to prove the high quality of the grown MAPbBr₃ crystal, we have performed high-resolution X-ray diffraction (HRXRD) on the three single crystal wafers. **Figure R4a** shows the rocking curves in a θ scan mode. We can see the MAPbBr₃ crystal wafers show an FWHM (full width at half-maximum) of 60.19 arcsec for (100), 40.28 arcsec for (110), and 86.72 arcsec for (111). As we all know that the FWHM of HRXRD or the X-ray diffraction rocking curve is an important index to evaluate crystalline quality of single crystals, wherein a perfect crystal produces a symmetrical and sharp peak profile with a small FWHM. Here our measured FWHM are much smaller than the reported values for MAPbBr₃ crystals¹⁻³ (**Figure R5a**), and is even comparable with those of the well-developed inorganic crystals⁴ (**Figure R5b**), demonstrating the grown crystals possessing high crystalline perfection. Furthermore, we have also measured the X-ray Laue diffraction patterns of the MAPbBr₃ wafers. As shown in **Figure R4b**, the well-defined laue diffractions further confirm the high crystal quality of the MAPbBr₃ wafers.

Figure R4. Crystalline quality characterizations of MAPbBr₃ wafers. (a) High-resolution X-ray diffraction rocking curves of MAPbBr₃ wafers on their respective (100), (110), and (111) planes. **(b)** Laue diffractions of MAPbBr₃ wafers on their respective (100), (110) and (111) planes.

Figure R5. High-resolution X-ray diffraction rocking curves of the reported MAPbBr₃ crystals¹ and the representative inorganic crystals⁴.

Steady-state and transient photoluminescence (PL) properties of MAPbBr₃ crystal were studied employing FLS-980 Edinburgh Instruments and the spectra were shown in **Figure R6**. The PL peak position of single crystal is located at 529 nm. The FWHM of PL of MAPbBr₃ single crystal is extremely narrow to 23 nm, indicating the grown MAPbBr₃ crystals possess high crystalline quality. **Figure R6b** shows the transient PL curve recorded at an excitation of 375 nm. The decay time fitted via tri-exponential showed three time components of a fast component (2.43 ns) and two slow dynamics (20.75 ns and 110.36 ns), which is consistent with some previous studies⁵⁻⁷.

Figure R6. Photoluminescence spectrum for MAPbBr₃ single crystal and PL time decay trace on a MAPbBr₃ single crystal at 529 nm through FLS-980 Edinburgh

Instruments.

The trap densities and carrier transport properties for different MAPbBr₃ wafers were evaluated by space charge limited current method. As shown in the **insets of Figure R7**, the two Au electrodes of the device were evaporated on the two opposite main surfaces for different wafers, thus leading to the carrier transport to be along different crystallographic orientations. The typical dark current-voltage (I - V) curves for (100), (110) and (111) wafers are shown in **Figure R7**, which showed the Mott–Gurney’s power law dependence. For the devices on all the three kinds of wafers, with the increasing bias voltage, an Ohmic region at lower bias and a quadratic dependence following a space charge-limited current model at higher bias were clearly separated by a kink point which is defined as the trap-filled limit voltage (V_{TFL}). The trap density (n_{trap}) and carrier mobility (μ) can be calculated by formula (4-1)⁸ and (4-2)⁹:

$$V_{TFL} = \frac{en_{trap}L^2}{2\varepsilon\varepsilon_0} \quad (4-1)$$

$$J_D = \frac{9\varepsilon\varepsilon_0\mu V_b^2}{8L^3} \quad (4-2)$$

Where e is the elemental charge (1.6×10^{-19} C), L is the wafer thickness, ε is the relative dielectric constant, ε_0 is the vacuum dielectric constant, J_D is the current density, and V_b is the applied voltage.

The typical trap density (n_{trap}) and carrier mobility (μ) derived from the I - V traces are in the orders of 10^{10} cm⁻³ and tens of cm² V⁻¹ S⁻¹, respectively. Both μ and n_{trap} of our crystal wafers are comparable to the values of bulk single crystals reported by other researchers¹⁰ and are much superior than those of polycrystalline films¹¹.

When compared among the different oriented wafers, we found the trap densities and carrier mobilities indeed vary with different orientations. As shown in **Figure R8**, the histogram statistics of n_{trap} and μ plotted from ten devices for each wafers, the trap densities are $2.52 \pm 1.64 \times 10^{10}$ cm⁻³, $2.86 \pm 2.38 \times 10^{10}$ cm⁻³ and $6.97 \pm 1.64 \times 10^{10}$ cm⁻³, for the (100), (110) and (111) wafers, respectively; the carrier mobilities are 80.54 ± 31.40 cm² V⁻¹ S⁻¹, 72.34 ± 17.19 cm² V⁻¹ S⁻¹ and 57.68 ± 14.53 cm² V⁻¹ S⁻¹, for the (100), (110) and (111) wafers, respectively. We can see the (100) wafers show the lowest trap density and the highest carrier mobility; while the (111) wafers show the highest trap density and the lowest carrier mobility. Thus, the defect density and migration behaviors along different crystallographic directions in MAPbBr₃ single crystal seem to be orientation-dependent.

Figure R7. Typical dark current–voltage curves of the space charge-limited current (SCLC) devices on (100), (110) and (111) MAPbBr₃ crystal wafers. Insets in respective panels show charge transport directions along different crystallographic orientations.

Figure R8. Diagrams of the distribution of derived trap densities and carrier mobilities depending on different wafer orientations.

The above discussions are added into the main text and the supporting information file, and are highlighted in red, and Figure R4 is added as Figure S1.

References

- 1 Jing L, Cheng X, Yuan Y, et al. Design growth of triangular pyramid MAPbBr₃ single crystal and its photoelectric anisotropy between (100) and (111) facets [J]. *The Journal of Physical Chemistry C*, **2019**, 123(17): 10826-10830.
- 2 Su J, Sang L, Wang D, et al. Solution growth and morphology of CH₃NH₃PbBr₃ single crystals in different solvents [J]. *Crystal Research and Technology*, **2016**, 51(11): 650-655.
- 3 Peng W, Wang L, Murali B, et al. Solution-grown monocrystalline hybrid perovskite films for hole-transporter-free solar cells [J]. *Advanced Materials*, **2016**, 28(17): 3383-3390.
- 4 Hu Q, Jia Z, Volpi A, et al. Crystal growth and spectral broadening of a promising Yb:CaLu_xGd_{1-x}AlO₄ disordered crystal for ultrafast laser application [J]. *CrystEngComm*, **2017**, 19(12): 1643-1647.

- 5 Wang K H, Li L C, Shellaiah M, et al. Structural and photophysical properties of methylammonium lead tribromide (MAPbBr₃) single crystals [J]. *Scientific reports*, **2017**, 7(1): 13643.
- 6 Yamada T, Yamada Y, Nishimura H, et al. Fast free-carrier diffusion in CH₃NH₃PbBr₃ single crystals revealed by time-resolved one-and two-photon excitation photoluminescence spectroscopy [J]. *Advanced Electronic Materials*, **2016**, 2(3): 1500290.
- 7 Eperon G E, Habisreutinger S N, Leijtens T, et al. The importance of moisture in hybrid lead halide perovskite thin film fabrication [J]. *ACS nano*, **2015**, 9(9): 9380-9393.
- 8 Saidaminov M I, Abdelhady A L, Murali B, et al. High-quality bulk hybrid perovskite single crystals within minutes by inverse temperature crystallization [J]. *Nature communications*, **2015**, 6(1): 7586.
- 9 Han Q, Bae S H, Sun P, et al. Single crystal formamidinium lead iodide (FAPbI₃): insight into the structural, optical, and electrical properties [J]. *Advanced Materials*, **2016**, 28(11): 2253-2258.
- 10 Liu Y, Yang Z, Liu S. Recent progress in single-crystalline perovskite research including crystal preparation, property evaluation, and applications [J]. *Advanced Science*, **2018**, 5(1): 1700471.
- 11 Saidaminov M I, Adinolfi V, Comin R, et al. Planar-integrated single-crystalline perovskite photodetectors [J]. *Nature communications*, **2015**, 6(1): 8724.

6. Page 13: line 291-292; *The defect-related deep traps would be healed by the trap passivation originating from oxygen/water Diffusion. How is the oxygen/water diffusion confirmed? Provide the XPS spectra before and after laser processing?*

Authors reply: Thanks for the constructive suggestion. According to the suggestion, we performed the micro-PL spectra measurements and X-ray photoelectron spectroscopy (XPS) tests of the MAPbBr₃ crystal wafers before and after laser processing, respectively, in order to better elucidate the effect of oxygen/water on the fluorescence enhancement.

As shown in **Figure R9**, compared to the PL of the unprocessed area on the MAPbBr₃ crystal, the PL intensity of the processed area is improved about 3 orders of magnitude in the air environment. In contrast, in the nitrogen environment, the PL intensity of the processed area is significantly weaker than that in the air under the same test conditions, nevertheless, still showing an enhancement of about 2 orders of magnitude compared to that of the unprocessed regions. In addition, in nitrogen the PL wavelength is blue shifted compared with that in air. This indicates that laser processing indeed result in fluorescence enhancement, wherein the limited carrier diffusion scale in the laser-induced microstructures would facilitate radiative recombination; and moreover, exposure to oxygen in air would further boost the photoluminescence of the microstructures. Similar emission enhancements caused by oxygen exposure have been reported in nano-sized

metal-halide perovskites, such as CsPbBr₃ nanowires.¹ In these cases, the passivation of deep-level trap states caused by excess lead atoms on the perovskite surface was identified as the responsible mechanism. Herein we believe the laser-induced microstructures with much larger specific surface area would facilitate oxygen diffusion and passivation of the non-radiative defect-related deep traps.

Then, we further analysed the X-ray photoelectron spectroscopy (XPS) of MAPbBr₃ crystals before and after laser processing. **Figure R10** shows the Pb 4f spectra and the corresponding analytical fitting. For the pristine MAPbBr₃ (unprocessed crystals), the Pb 4f peaks were deconvolved into two components: the large higher binding energy peaks corresponding to Pb 4f levels of MAPbBr₃ (138.4 eV for 4f_{7/2} and 143.3 eV for 4f_{5/2}),² and the associated smaller peaks with binding energy 136.5 eV and 141.4 eV, which can be assigned to metallic Pb.³ This indicates that although the grown MAPbBr₃ crystals possess a very high crystalline perfection, excess Pb atoms can still emerge in the crystal, possibly due to the unintended losses of Br atoms or incomplete reaction between MABr and PbBr₂.^{2,3} It has been shown that such metallic Pb atoms would cause deep impurity levels in the band gap, which act as nonradiative decay channels to degrade the photoluminescence.¹ While for the processed MAPbBr₃ crystal, the XPS analysis reveals that the metallic Pb peaks decreased dramatically; and besides the higher binding energy peaks corresponding to Pb 4f levels of MAPbBr₃ (138.4 eV for 4f_{7/2} and 143.3 eV for 4f_{5/2}), a set of lower binding energy peaks at 138.2 eV and 143.1 eV that can be assigned to Pb coordination with adsorbed oxygen (denoted as O_{passivated-Pb}) became remarkable.⁴ Thus, the passivation of the deep level trap state was achieved through oxygen binding to the metallic lead defects.

In conclusion, these results demonstrate that laser processing can dramatically boost the photoluminescence of MAPbBr₃ crystal, wherein oxygen exposure passivation plays a key role. Femtosecond lasers on the one hand produced microstructures with nanoscale thickness, which limit free carrier diffusion and facilitate radiative recombination, and on the other hand, the tentacle-like microstructures with a large specific surface area increase exposure to oxygen and facilitate oxygen diffusion, which passivates the deep level trap state caused by the metallic Pb atoms. XPS analysis of the MAPbBr₃ crystal wafers before and after laser processing confirms oxygen binding to the metallic Pb defects.

Figure R9. Micro-PL spectra of the unprocessed MAPbBr₃ crystal in air (black line), laser-processed MAPbBr₃ crystal in nitrogen (blue line), and the laser-processed MAPbBr₃ crystal in air (red line). A continuous-wave (CW) laser beam at 409 nm as excitation wavelength was focused onto the sample surface by a 10×objective (Olympus, NA 0.25).

Figure R10. Binding energy spectra and the corresponding analytical fitting for MAPbBr₃ crystals before and after laser processing.

The above discussions have been added in the main text and are highlighted in red. Figure R10 is added into Figure 4 of the main text.

References

- 1 Lu D, Zhang Y, Lai M, et al. Giant light-emission enhancement in lead halide perovskites by surface oxygen passivation [J]. *Nano letters*, **2018**, 18(11): 6967-6973.
- 2 Lindblad R, Jena N K, Philippe B, et al. Electronic structure of CH₃NH₃PbX₃ perovskites: dependence on the halide moiety [J]. *The Journal of Physical Chemistry C*, **2015**, 119(4): 1818-1825.
- 3 Cho H, Jeong S H, Park M H, et al. Overcoming the electroluminescence efficiency limitations of perovskite light-emitting diodes [J]. *Science*, **2015**, 350(6265): 1222-1225.
- 4 Li D, Meng Y, Zhang Y, et al. Selective Surface Engineering of Perovskite Microwire Arrays [J]. *Advanced Functional Materials*, **2023**: 2302866.

7. Additionally, please address any limitations of the study and suggest directions for future research and highlight the significance of the findings.

Authors reply: Thanks for the constructive suggestion. According to the suggestion, we have made a more comprehensive analysis and discussion about the limitations, significance and perspective of this study, as follows:

Based on the differently-oriented MAPbBr₃ single-crystal wafers, this study presents innovative findings primarily in two aspects. Firstly, utilizing polarization-dependent ultrafast time-resolved spectroscopy, the anisotropic dynamics of photo-excited carriers on the picosecond time scale was firstly revealed. This discovery provides a deeper understanding of the ultrafast carrier relaxation pathways from a perspective of crystal orientation. It holds significant implications for exploring and expanding the potential of perovskite single crystals in polarization-sensitive and ultrafast optoelectronics applications, such as optical modulators, high-speed light polarization sensors, and ballistic transistors, which require both polarization sensitivity and high-field running capability simultaneously. However, a comprehensive understanding to correlate the observed polarization-dependent dynamics with the crystallographic structures has not been achieved yet due to limitations in current excited-state experimental techniques. Further progress will rely on employing more advanced ultrafast probing techniques, in combination with theoretical simulations, to comprehensively elucidate the observed carrier dynamics behind the underlying excited structure.

Secondly, by employing femtosecond laser processing, luminescent patterns with a

remarkable three-order-of-magnitude PL enhancement on the bulk single crystals were achieved. The observed enhancement can be ultimately attributed to the synergy of three factors: the limited carrier diffusion length, the increase in shallow trap-assisted recombination centers, and the passivation of deep traps within the femtosecond laser-induced tentacle-like microstructures. In addition to offering a convenient top-down strategy for enhancing the photoluminescence intensity of bulk crystals, this study has also provided an in-depth understanding of the luminescence mechanism from multiple spatial (bulk and micro/nano-scale) and temporal (steady and transient-state) dimensions. Considering the delicate relationship between the trap states and the photoluminescence capacity of the 3D perovskite single crystals, looking ahead, further research can delve into the origin of surface defect states to gain a deeper understanding of the luminescence mechanism.

The above content has been added to the **Discussion Section** of the main text and highlighted in red.

Response to the Comments and Suggestions of Reviewer 2

The reviewer's comments: *Ge et al report on two independent observations on oriented single crystals of methylammonium lead bromide: one on the anisotropy of transient dynamics and on the effect of laser processing on the photoluminescence efficiency. While the presented observations are interesting, I have several concerns with the presented conclusions and thus cannot recommend publication in current form.*

Authors reply: We sincerely thank the reviewer for his/her attention and interest in our work, and put forward constructive suggestions for us to improve the quality of manuscripts.

1. *It is critical to note that the single crystals are semiconductors with well defined band structure and the representation in Figure 2e is incorrect. The photoexcitation creates a population of electrons (holes) in the conduction (valence) band, which thermalize to the bottom of the band in a few hundreds of femtoseconds through phonon scattering. The longer time dynamics are a results of recombination, through both radiative and non-radiative (primarily defects) channels. The observed anisotropy has to be discussed within this well established photophysical scenario. The absence of anisotropy in the (110) crystal must be properly discussed.*

Authors reply: Thanks for the reviewer's constructive suggestions and we agree with the comment that the original representation in Figure 2e is not proper. According to the suggestion, we have revised the schematic diagram of the photoexcited carrier relaxation dynamics in the form of energy band structure, and improved the discussions about the anisotropic carrier dynamics following the above bandgap pulsed laser excitation of MAPbBr₃.

As shown in **Figure R11**, the initial photoexcitation results in the generation of a population of hot carriers (HCs) through the absorption of high-energy photons (above the bandgap). After that, the HCs will firstly redistribute the energy among themselves through elastic carrier-carrier interactions and scattering, to a Fermi-Dirac distribution with a temperature greater than the lattice temperature. This process occurs very rapidly (normally within 100 fs) and is usually referred to as carrier thermalization. Following carrier thermalization, the thermalized HCs lose their excess energies and equilibrate with the crystal lattice mainly through inelastic scattering events. Here it is noted that the former stage immediately following photoexcitation is referred to as carrier thermalization while the latter process as HCs cooling.

In this experiment, our main focus is on HCs cooling and the subsequent dynamics of "cold" carriers near the band-edge. In a recent review by Tze Chien Sum et al. [*Chem. Rev.*

2023, 123 (13), 8154], it is stated that the HCs cooling regime typically occurs within a timescale of up to ~ 10 ps in halide perovskites, where HCs lose their excess energies and relax down to the band-edge. The dynamics involved in this regime include HC cooling, HC trapping, polaron formation, and exciton formation.¹ The large exciton binding energy in MAPbBr₃ enables the formation of excitons, coexisting with free carriers. Due to the polar nature of the MAPbBr₃ crystals in this experiment, the dominant relaxation pathway for the HCs in MAPbBr₃ is through Fröhlich interactions, i.e., carrier–LO–phonon scattering within 1~2 ps (**Stage I**). Large polarons can be formed due to the prominent carrier–phonon interactions coupled with lattice deformations.² Then the emitted LO phonons from carrier–LO–phonon scattering would decay into the daughter acoustic phonons (corresponding to τ_1 in this experiment, **Stage II**). Following this regime, these HCs have cooled almost to the band-edge. In next regime, the energy will continue to be lost via decay of acoustic phonon emissions, which usually occurs in hundreds of picoseconds³ (corresponding to τ_2 in this experiment, **Stage III**), involving the dynamics of thermalized “cold” carriers near the band-edge before their eventual trapping and recombination. Photoinduced lattice expansion, stress, strain, and coherent phonon effects are evident in this regime. Lastly, the cooled carriers will recombine (usually in nanosecond timescale).

Figure R11. Schematic of the photoexcited carrier relaxation dynamics: (Stage I) electron-LO-phonon scattering (Fröhlich interaction); (Stage II) optical phonon decay to acoustic phonons; (Stage III) acoustic phonons decay and propagation.

In the current experiment, there exist three distinct decay regimes: (1) electron–LO–phonon scattering (Fröhlich interaction); (2) LO–phonon decay to acoustic phonons; (3) acoustic phonons decay and propagation. The decay time scale mainly concerns the carrier

dynamics involving hot carrier cooling, polaron formation, exciton formation, and the subsequent dynamics of “cold” carriers near the band-edge involving photoinduced lattice expansion, strain, and coherent phonon effects. During these photophysical processes, the phonons play a significant role in local lattice displacements which generate a polarization-induced electric field that in turn interacts with the charge carriers. Thus, the involvement of phonons (lattice vibrations) in all these stages correlates the carrier relaxation dynamics with the lattice structure.

From a structural view, although MAPbBr₃ is regarded as a highly symmetric cubic structure at room temperature resting on identification of the organic MA⁺ cation as having random orientations in the X-ray crystallography statistics,⁴ our group and other researchers have indeed revealed in-plane and out-of-plane structure anisotropy through angle-resolved Raman spectroscopy, as well as orientation-dependent optoelectronic properties by using photoconductivity and photoluminescence measurements.⁵⁻⁸ These recent investigations unambiguously corroborated the steady-state anisotropy on MAPbBr₃ single crystals, however, the transient-state anisotropy in the ultrafast carrier dynamics still remains sealed. This is because the anisotropy of steady-state properties relies directly on the static crystal structure that we can depict intuitively; while for the ultrafast carrier dynamics occurring in the excited state, there are two main obstacles. Firstly, photoexcitation can lead to a distinct excited electronic structure compared to the ground state (so far, there is still a big challenge to accurately capture the excited electronic structures experimentally). Secondly, the carrier relaxation process is heavily influenced by phonons, which are hard to describe with a clear picture. Therefore, it is infeasible to directly correlate the carrier relaxation dynamics with the electronic structure of the ground state. In spite of these challenges, we can still find clues from analyzing the distortion of the PbBr₆ octahedral framework and the dynamic orientation of MA⁺ cations. Additionally, investigating the defect density distribution along different crystallographic orientations and considering the surface and many-body effects can provide further understanding. These factors may affect the excited electronic structure and phonon behavior of MAPbBr₃ single crystal.

Lattice deformation, vibration and relaxation effect: Because of the different crystallographic geometric configurations of the (100), (110) and (111) planes (**Figure R1**, including the atomic density, interplanar spacing, etc.) and the distribution of MA⁺ orientation domains, the equilibrium out-of-plane and in-plane anisotropic structure should induce non-synchronized lattice deformation and relaxation upon photoexcitation, which may break the original lattice symmetry in the excited state structure. E.g., the MA⁺ rocking and twisting vibration modes demonstrate distinct in-plane anisotropy on (100) and (110) wafers, which may in turn cause interactions on the distortion of the PbBr₆⁴⁻ inorganic skeletons; and the transient photoluminescence lifetimes also show discrepancies among (100), (110) and (111) wafers.⁵ Consequently the incoherent lattice vibration amplitude and

frequency dictate the specific phonon modes along different crystallographic orientations in MAPbBr₃. Thus an azimuthally balanced lattice deformation on the (110) plane is imagined to induce a polarization-independent excited state dynamics on the (110) crystal plane.

Trap density distribution and surface effect: Because on the different crystallographic planes, the defect formation energies of vacancies and interstitials are different; and likewise, the ion migration also depends on the crystallographic orientations, these will lead trap density distributions inhomogeneously on different crystal wafers. In **Figure R7 & R8**, we have demonstrated the discrepancy of trap densities on differently oriented wafers. Carrier scattering and trapping resulting from defects play a crucial role in influencing carrier relaxation in different orientations within the bulk crystal. Additionally, the polaronic nature of the relatively soft lattice in MAPbBr₃ introduces surface dipoles that affect electronic structure and charge distribution. These factors also become significant determinants of carrier transport and recombination properties on distinct crystal planes. Thus a more highly symmetrical distribution of the excited-state charge density on the (110) crystal plane is anticipated.

Many-body effect: The interactions among electrons and with other particles, such as phonons (lattice vibrations) and other carriers, can lead to many-body effects. These effects cause a redistribution of momentum and energy of the excited state electrons through energy exchange or resonance coupling between the particles. Many-body effect has significant consequences on the carrier's lifetime and optical properties such as absorption and refraction in the excited state, thus making the carrier dynamics even more complicated.

Therefore, due to the uncomparable excited state electronic structure with that of the ground state, the mechanisms behind ultrafast hot carrier relaxation process are too subtle to be accurately learned. Frankly speaking, we still do not have exact answers about why MAPbBr₃ (110) wafer behaves distinctly in the excited-state carrier relaxations compared to other wafers. It is important to note that it requires more detailed experimental investigation and deeper analysis to correlate the observed polarization-dependent dynamics with the crystallographic structures in different crystal planes. This study, based on the differently-oriented MAPbBr₃ single-crystal wafers, firstly penetrated into the orientation-dependent dynamical evolution of the excited carriers and provided solid observations of the anisotropy in the ultrafast carrier dynamics. Although a comprehensive understanding has not been achieved yet due to limitations in excited-state experimental techniques, this discovery holds significant implications as it provides a novel perspective for a deeper understanding of the ultrafast carrier relaxation pathways and opens up new possibilities for utilizing perovskite single crystals in polarization-sensitive photoelectron responses.

The above discussions are added to the main text and supporting information file, and are highlighted in red. Figure R11 is added into Figure 2 of the main text to replace the original Figure 2e.

References

- 1 Fu J, Ramesh S, Melvin Lim J W, et al. Carriers, Quasi-particles, and Collective Excitations in Halide Perovskites [J]. *Chemical Reviews*, **2023**, *123*(13): 8154-8231.
- 2 Miyata K, Meggiolaro D, Trinh M T, et al. Large polarons in lead halide perovskites [J]. *Science advances*, **2017**, *3*(8): e1701217.
- 3 Li M, Fu J, Xu Q, et al. Slow hot-carrier cooling in halide perovskites: prospects for hot-carrier solar cells [J]. *Advanced Materials*, **2019**, *31*(47): 1802486.
- 4 Nandi P, Pandey S K, Giri C, et al. Probing the electronic structure of hybrid perovskites in the orientationally disordered cubic phase [J]. *The Journal of Physical Chemistry Letters*, **2020**, *11*(14): 5719-5727.
- 5 Zhang L, Cui S, Guo Q, et al. Anisotropic Performance of High-Quality MAPbBr₃ Single-Crystal Wafers [J]. *ACS Applied Materials & Interfaces*, **2020**, *12*(46): 51616-51627.
- 6 Zhang L, Liu Y, Ye X, et al. Exploring anisotropy on oriented wafers of MAPbBr₃ crystals grown by controlled antisolvent diffusion [J]. *Crystal Growth & Design*, **2018**, *18*(11): 6652-6660.
- 7 Zuo Z, Ding J, Zhao Y, et al. Enhanced optoelectronic performance on the (110) lattice plane of an MAPbBr₃ single crystal [J]. *The Journal of Physical Chemistry Letters*, **2017**, *8*(3): 684-689.
- 8 Yang H, Zhou Y, Yang Y, et al. Crystal facet engineering induced anisotropic transport of charge carriers in a perovskite [J]. *Journal of Materials Chemistry C*, **2018**, *6*(43): 11707-11713.

2. *A very important aspect is what one is probing at 1.55 eV? From the TA spectra in figure 4, it appears that the authors are probing bleach dynamics of mid-gap states. How can anisotropy be explained for spectral features of defect states, that are localized and randomly spread over the bulk of the crystal? To be more precise, one must do the experiment at the bandedge.*

Authors reply: Thanks for the comment. As the reviewer has concluded above, in this experiment an important objective is to monitor the hot-carriers relaxation dynamics occurring in the excited states on differently orientated wafers. It has been reported that there should be two kinds of primary photoexcitations in the hybrid perovskites, namely free carriers in the valence band (VB) and/or conduction band (CB); and excitons which are Coulomb correlated electron-hole (e-h) bound pairs. The reported exciton binding energy of MAPbBr₃ perovskite ranges from 40 to 150 meV,¹⁻³ larger than the thermal energy (~26 meV) at room temperature. Therefore, the photogenerated excitons and free carriers are expected to coexist according to Saha–Langmuir equation.⁴⁻⁶ (“with typical

excitation densities of around 10^{16} – 10^{17} cm^{-3} for PL quenching experiments, free carrier population is found to be in the range of $\approx 50\%$ – 90% when trap states are not considered.” Tze Chien Sum et al., *Adv. Energy Mater.* **2016**, 1600551) Additionally, according to the reference suggested by the reviewer below (*Nat. Photonics* **2015**, 9, 695), it is reported that the electrostatic potential variations in smaller polycrystals suppress exciton formation, while the larger single-crystals of the same composition demonstrate an unambiguous excitonic state.

In our work the relaxation dynamics of the excited-carriers in the picoseconds timescale was monitored by using a polarized probe pulse at 1.55 eV, following an above-bandgap pump at 3.1 eV (the energy bandgap of MAPbBr₃ single crystal is ~ 2.3 eV) with low excitation density (1.4×10^{15} cm^{-3}). According to the retested TA spectra (**Figure R12a**), there is observed an obvious positive ΔA signal, signifying the occurrence of photoinduced absorption (PA) in the broad spectral region of $\lambda > 600$ nm (**Figure R12b**). The PA signal became stronger as the excitation density increased (**Figure R12c**), which confirms that it should mainly originate from the excited hot carriers. Here, we ascribe the negative ΔA signals close to the optical gap as the ground-state bleaching (GSB), which is attributed to state-filling by excited carriers. However, it is evident that the dynamic of PA is different from that of the bleaching band, as shown in the **Figure R12d**, indicating that these two bands originate from different photoexcitations. We therefore assign the PA band to reflect dynamics of the photogenerated excitons. The observation is consistent with the investigation by Anita Ho-Baillie et al. [*J. Phys. Chem. C* **2016**, 120, 2542] that “The much faster rise of PA₂ indicates a different origin from bleaching, which is most likely to be the absorption of photogenerated excitons for several reasons.” (**Figure R13a**) Sheng et al. [*PRL* **2015** 114, 116601] also attributed a similar absorption band after photoexcitation of MAPbPbI₃ to generation of exciton (**Figure R13b**).

Figure R12. TA spectra of MAPbBr₃ crystal. (a) Pseudo colour TA plot with the excitation density of 1.45×10^{16} cm^{-3} . (b) TA spectrum at different delay time. (c) TA spectrum at different excitation densities at 1.5 ps. (d) The ground-state bleaching (GSB) and photoinduced absorption (PA) dynamics are probed at 555 nm and 800 nm, respectively.

Figure R13. TA spectra with exciton characteristics. (a) TA spectra of MAPbBr₃ film showing a long wavelength PA band (PA₂).⁷ **(b)** TA spectra and the decay dynamics of the photoinduced bleaching (PB) and a PA band of MAPbI₃ film. Note that the dynamic evolution of PA₁ was obtained by probing at a low-energy spectral range.⁸

Generally, the optical spectrum signature of free carrier absorption (FCA) follows the Drude model, which can be described by the relation: $FCA \sim N/[1+(\omega\tau_s)^2]$,⁹ where N is the photocarriers concentration, τ_s is the momentum scattering time, and ω is the photon angular frequency. Therefore the FCA contribution to PA is limited to frequencies. The FCA spectrum in perovskites compounds would be relevant only in the spectral range of few tens of meV.¹⁰ In contrast to free carriers, excitons may have PA bands in the long-wavelength range extending to the mid-infrared region that originate from excitonic intersub-band or/and interband transitions. Thus, by probing the dynamics at 1.55 eV (located at the PA band), one may hit the aim to monitor the relaxation of photogenerated excitons.

It should be pointed out that probing the PA band dynamics would not affect the judgment of the in-plane and out-of-plane carrier dynamics anisotropy among different orientations wafers. As Zeev Valy Vardeny et al. [*Adv. Funct. Mater.* **2016**, *26*, 1617] found that both excitons and free carriers PA bands in MAPbI₃ exhibit transient photoinduced polarization memory, which originates from the crystal anisotropy of this hybrid perovskite.

As to how correlating anisotropy with the defect states, we agree with the reviewer that through ultrafast carrier relaxation dynamics spectroscopy, it is unlikely to illuminate the anisotropic distribution of defect states among differently oriented wafers. Because the defect states are localized randomly throughout the bulk of the crystal and the transmission probing mode we adopted in this study, we did not aim to manifest the presence of defect states or investigate their in-plane distribution among different wafers by the ultrafast dynamic spectra. Alternatively, by using space charge limited current (SCLC) measurements performed on (100), (110) and (111) wafers, we could assess the

approximate distribution of trap densities among different crystal orientations. As shown in **Figure R7 & R8**, the measured trap densities indeed show orientation-dependence along different crystallographic directions in MAPbBr₃ single crystal.

The above discussions have been added to the main text and supporting information file, and are highlighted in red. Figure R12 is added as Figure S10.

References

- 1 Tanaka K, Takahashi T, Ban T, et al. Comparative study on the excitons in lead-halide-based perovskite-type crystals CH₃NH₃PbBr₃ CH₃NH₃PbI₃ [J]. *Solid state communications*, **2003**, 127(9-10): 619-623.
 - 2 Koutselas I B, Ducasse L, Papavassiliou G C. Electronic properties of three- and low-dimensional semiconducting materials with Pb halide and Sn halide units [J]. *Journal of Physics: Condensed Matter*, **1996**, 8(9): 1217.
 - 3 Yang Y, Yang M, Li Z, et al. Comparison of recombination dynamics in CH₃NH₃PbBr₃ and CH₃NH₃PbI₃ perovskite films: influence of exciton binding energy [J]. *The journal of physical chemistry letters*, **2015**, 6(23): 4688-4692.
 - 4 Sestu N, Cadelano M, Sarritzu V, et al. Absorption F-sum rule for the exciton binding energy in methylammonium lead halide perovskites [J]. *The journal of physical chemistry letters*, **2015**, 6(22): 4566-4572.
 - 5 Saba M, Cadelano M, Marongiu D, et al. Correlated electron-hole plasma in organometal perovskites [J]. *Nature communications*, **2014**, 5(1): 5049.
 - 6 D'innocenzo V, Grancini G, Alcocer M J P, et al. Excitons versus free charges in organo-lead tri-halide perovskites [J]. *Nature communications*, **2014**, 5(1): 3586.
 - 7 Deng X, Wen X, Huang S, et al. Ultrafast carrier dynamics in methylammonium lead bromide perovskite [J]. *The Journal of Physical Chemistry C*, **2016**, 120(5): 2542-2547.
 - 8 Sheng C X, Zhang C, Zhai Y, et al. Exciton versus free carrier photogeneration in organometal trihalide perovskites probed by broadband ultrafast polarization memory dynamics [J]. *Physical review letters*, **2015**, 114(11): 116601.
 - 9 J. I. Pankove, *Optical Processes in semiconductors*, Dover Books, Dover, NY **1971**.
 - 10 Zhai Y, Sheng C X, Zhang C, et al. Ultrafast spectroscopy of photoexcitations in organometal trihalide perovskites [J]. *Advanced Functional Materials*, **2016**, 26(10): 1617-1627.
3. *What is effect of the femtosecond laser in the second part of the manuscript? It appears that the top part of the crystal is getting ablated by the laser (through two photon absorption as the authors point out). It is known that the surface of bulk perovskite crystals are intrinsically defective and by ablation, one is revealing the lesser defective bulk phase of the crystal. See Applied Physics Reviews, 6, 031401.*

Authors reply: We sincerely thank the reviewer for the constructive comment. As mentioned by the reviewer, by utilizing the two-photon absorption effect, the femtosecond

laser can ablate specific regions of the crystal surface, leading to the formation of a mass of micro-/nano-scaled structures along the scanned region. This process ultimately achieves a remarkable three-order-of-magnitude enhancement in photoluminescence (PL). We attribute the main effects of the femtosecond laser on the enhancement of PL to two factors. One is to induce tentacle-like micro-/nano-structures, which restricts the diffusion range of free carriers, thus prohibiting the long-range diffusion inward to the nonradiative bulk region. The other is to induce abundant shallow trap-assisted recombination centers. Furthermore, considering the large specific area surfaces of the formed micro-/nano-structures, the presence of ambient oxygen leads to effective passivation of nonradiative deep-level traps, transforming them into radiative shallow-traps.

It has been reported that the trap state density at the surface of the halide perovskite single crystals is indeed much higher compared to the bulk. (*“The trap density near the interface region was ~10-fold greater than that inside the single crystal. This difference indicated that dangling bonds at the surface of the crystal form charge traps.”* Huang et al., *Science* **2020**, 367, 1352) However, in this study, if only ablating the surface portion of the crystal by femtosecond laser, the exposed bulk phase still displays no photoluminescence, as shown in **Figure R14**. It is when the femtosecond laser processing duration extending to the appearance of tentacle-like microstructures, green emissions emerge concomitantly. Subsequently, we tried to provide proofs for the effect of laser processing on the effective carrier recombination. Overall, the formation of micro-/nano-structures with limited carrier diffusion range, and the creation of shallow-trap states induced by femtosecond laser and/or transformed through the nonradiative deep-level traps passivated by ambient oxygen are considered the key factors responsible for the observed enhancement in PL. As discussed in the manuscript, the shallow trap-assisted luminescence centers coincide with the bathochromic shift in the emission wavelength and the broad exciton transient absorption band, compared to that of the pristine bulk crystal, as depicted in the PL spectra and the TAS spectra, respectively.

Figure R14. The impact of femtosecond laser processing parameters on morphology and photoluminescence of MAPbBr₃.

In the reviewer suggested literature, Duim et al. reported the surface defect passivation of bulk MAPbBr₃ single crystals by employing benzylamine. Benzylamine can replace the methylammonium cations in MAPbBr₃ and drive the etching of the crystal surface. Initially, square etch pits are formed and scattered on the surface, eventually merging into large rectangular terraces. They found that the surface defect passivation leads to an enhancement of the photoluminescence intensity by over two orders of magnitude. However, when compared to the laser-processed patterns as reported in this manuscript, the passivated surface is still too weak to be perceived by the naked eyes.

We thank the reviewer for providing us the useful and relevant article. The original report (Herman Duim, Hong-Hua Fang, Sampson Adjokatse, Gert H. ten Brink, Miguel A. L. Marques, Bart J. Kooi, Graeme R. Blake, Silvana Botti, and Maria A. Loi. Mechanism of surface passivation of methylammonium lead tribromide single crystals by benzylamine. *Appl. Phys. Rev.* **2019**, 6, 031401) is included in our revised manuscript as Reference 35.

The above discussions are added to the main text and are highlighted in red.

4. The TA analysis presented in figure 4 has a few issues.

- a. Firstly, if there is chirping (which looks negative?), it must be properly corrected. Moreover, I don't think the bleach is appearing later than the sub-gap states. The authors are creating a population way above in the CB and thermalization results in a few picoseconds in the appearance of the GSB at the bandedge (see *Nature Photonics*, 9, 695).

Authors reply: We sincerely thank the reviewer’s constructive suggestion. In the original TA spectrum (Figure 4 in the manuscript, represented here as **Figure R15a**), it indeed exhibits a chirp phenomenon due to the larger thickness of the bulk crystal. It can be observed that the lower frequency signals appear slightly delayed compared to the higher frequency signals in the TA spectrum, especially during the initial few picoseconds. This can be assigned to the positive dispersion of the white light continuum beam (red wavelengths travel faster than the blue ones), a phenomenon being commonly referred to as the positive chirp effect. Therefore, we have re-performed the transient absorption spectroscopy (TAS) measurements on the MAPbBr₃ wafer by using the ultrafast TA spectrometer (Harpia-TA, Light Conversion) and applied chirp correction to the data (**Figure R15b & c**).

Figure R15. Pseudo colour TA plot of MAPbBr₃ (100) wafer. The original TAS without chirp correction (a), and the supplemented TAS with chirp correction plotted by linear (b) and logarithmic (c) delay time.

We agree with the comment that the bleach would not appear later than the sub-gap states. Generally, the sub-gap transient absorption signal is attributed to the Stark effect, where photo-excited hot carriers generate localized fields that cause energy band restructuring through exchange or Coulomb interactions between carriers, resulting in the shift of the bandgap. The sub-bandgap state signal is typically observed as a derivative-like feature of the bleaching signal, located close to the bandgap, and generally occurs within a few hundred femtoseconds^{1,2} (**Figure R16a & b**). As mentioned in the literature recommended by the reviewer [*Nat. Photonics* **2015**, *9*, 695], the investigation of transient absorption (TA) spectra in MAPbBr₃ films with different morphologies and crystallinities revealed a rapid decay of the photoinduced absorption (PA) signal located close to the bandgap (sub-gap states), occurring in less than 1 ps (**Figure R16c**). In our experiment, a broad absorption band spanning the range of 600-900 nm was observed prior to the appearance of the bleach signal (**Figure R15c & R18e**), which does not conform to the characteristics of sub-gap states. This photoinduced absorption signal seems to be an inescapably coherent artifact that is present in bulk single crystals (with the thickness in

millimeters and above), which is attributed to the non-degenerate two-photon absorption (TPA) when the pump and probe beams overlap both temporally and spatially. (“*A small coherent artifact signal (occurring in subpicosecond timescale) is also observed when the pump and probe beam overlaps. Its mechanism is similar to that for 2PA excitation.*” Tze Chien Sum et al., *Adv. Energy Mater.* **2016**, 1600551).

We thank the reviewer for providing us this useful and relevant article. The original report (Giulia Grancini, Ajay Ram Srimath Kandada, Jarvist M. Frost, Alex J. Barker1, Michele De Bastiani, Marina Gandini, Sergio Marras, Guglielmo Lanzani, Aron Walsh and Annamaria Petrozza, Role of microstructure in the electron–hole interaction of hybrid lead halide perovskites. *Nat. Photonics* **2015**, 9, 695) is included in our revised manuscript as Reference 23.

The above discussions are added to the main text and supporting information file, and are highlighted in red. Figure R15b is added into Figure 4 of the main text to replace the original Figure 4g. Figure R15c is added as Figure S8.

Figure R16. Different excited state signal characteristics in TAS. (a) The TAS of MAPbBr₃ polycrystalline films at different excitation density.¹ **(b)** The sub-gap transient absorption signal characteristics.² **(c)** Photo-induced excited population of MAPbBr₃ as a function of crystal size.³

References

- 1 Li M, Bhaumik S, Goh T W, et al. Slow cooling and highly efficient extraction of hot carriers in colloidal perovskite nanocrystals [J]. *Nature communications*, **2017**, 8(1): 14350.

- 2 Price M B, Butkus J, Jellicoe T C, et al. Hot-carrier cooling and photoinduced refractive index changes in organic–inorganic lead halide perovskites [J]. *Nature communications*, **2015**, 6(1): 8420.
- 3 Grancini G, Srimath Kandada A R, Frost J M, et al. Role of microstructure in the electron–hole interaction of hybrid lead halide perovskites [J]. *Nature photonics*, **2015**, 9(10): 695–701.

b. *The narrow spectral feature at around 703 nm in Figure 4h appears to be an artefact, most likely from the saturation of the detector. The authors should show the transmission spectrum of the white light taken at the same condition as the measurement to discount this effect.*

Authors reply: Thanks for the reviewer’s constructive suggestions. According to the suggestion, we have re-performed the transient absorption spectroscopy (TAS) measurements on the MAPbBr₃ wafer by using a commercial ultrafast TA spectrometer (Harpia-TA, Light Conversion). We agree with the comment that the narrow spectral feature appeared in the original Figure 4h is an artefact. As shown in **Figure R15b & c**, there is no negative absorption feature at around 703 nm. Here the probe beam is a low-intensity continuous white light generated by focusing the output beam (1030 nm, 54 kHz, light conversion Ltd.) of the Yb:KGW laser system onto a 5mm sapphire crystal, ensuring high stability and reliability. **Figure R17** shows the transmission spectrum of the continuous white light (without pumping excitation) before and after passing through the sample under the same conditions as in the measurements. There is no apparent supersaturation around 703 nm in the transmission spectrum of the white light, and the sample does not demonstrate any notable absorption or emission at this wavelength. Thus, we speculate that the original negative absorption feature at around 703 nm in Figure 4h is due to the over-saturation of the detector.

Figure R17. The transmission spectrum of continuous white light (without pumping excitation) before and after passing through the MAPbBr₃ single crystalline wafer.

Accordingly, we have revised the TA spectra in the manuscript. As a result, we have discarded the previous assertion of the existence of trap states near 703 nm in the corresponding discussions.

c. There is no convincing data to substantiate the model in figure 4j and n. Intensity dependent dynamics should help here. However, looking at the TA intensity, the authors are likely to be in the high density Auger regime and the dynamics is not going to representative of the defect dynamics.

Authors reply: Thanks for the reviewer's constructive suggestions. In order to provide more comprehensive evidences to support the model depicted in Figure 4j and n (also represented here as **Figure R18g & h**), we conducted additional X-ray photoelectron spectroscopy (XPS) measurements on both pristine and laser-processed MAPbBr₃ crystals, in conjunction with photoluminescence (PL) spectroscopy and transient absorption spectroscopy (TAS). Through these combined experimental techniques and comparative studies, we were able to provide a detailed understanding of the substantial enhancement in photoluminescence upon laser processing.

Figure R18a & b show the Pb 4f spectra and the corresponding analytical fitting. For the pristine MAPbBr₃ (unprocessed crystals), the Pb 4f peaks were deconvolved into two components: the large higher binding energy peaks corresponding to Pb 4f levels of MAPbBr₃ (138.4 eV for 4f_{7/2} and 143.3 eV for 4f_{5/2}),¹ and the associated smaller peaks with binding energy 136.5 eV and 141.4 eV, which can be assigned to metallic Pb.² This indicates that although the grown MAPbBr₃ crystals possess a very high crystalline perfection, excess Pb atoms can still emerge in the crystal, possibly due to the unintended losses of Br atoms or incomplete reaction between MABr and PbBr₂.^{1,2} It has been proved that such metallic Pb atoms would cause deep defect levels in the band gap, which act as nonradiative decay channels to degrade the photoluminescence,³ as shown in **Figure R19a**. While for the processed MAPbBr₃ crystal, the XPS analysis reveals that the metallic Pb peaks decreased dramatically; and besides the higher binding energy peaks corresponding to Pb 4f levels of MAPbBr₃ (138.4 eV for 4f_{7/2} and 143.3 eV for 4f_{5/2}), a set of lower binding energy peaks at 138.2 eV and 143.1 eV that can be assigned to Pb coordination with adsorbed oxygen (denoted as O_{passivated}-Pb) became remarkable.⁴ It is because of the large specific area surfaces of the micro-/nano-structures formed during laser processing,

effective passivation by ambient air took place through oxygen binding to the defects of metallic Pb, leading to the nonradiative deep-level traps transforming into radiative shallow-traps.

Figure R18. The analysis of the PL mechanism of bulk crystal and microstructure of MAPbBr₃. (a-b) Binding energy spectra and the corresponding analytical fitting for MAPbBr₃ crystals before and after laser processing. (c-d) Pseudo colour TA plots before and after laser processing with a low excitation density of $1.45 \times 10^{16} \text{ cm}^{-3}$. (e-f) TA spectra before and after laser processing. (g-h) Flat-band energy level diagram for illustration of different relaxation pathways of the hot-electron in unprocessed (g) and processed (h) MAPbBr₃ crystals.

In the photoluminescence spectra (**Figure 19b**), we observed not only a significant enhancement in emission intensity but also a noticeable redshift after femtosecond laser processing. This can be attributed mainly to two factors. Firstly, the femtosecond laser processing introduces shallow defects level near the band edge, which can act as beneficial recombination centres to enhance the emission. Secondly, the redshift and enhanced emission can be attributed to an increase in the population of band edge excitons owing to the size confinement effect of the microstructure. Furthermore, we performed transient absorption spectroscopy (TAS) on both pristine and laser-processed MAPbBr₃ wafers under identical conditions, as shown in **Figure R18c-f & R19c**. We used an excitation of 3.1 eV with a low excitation density of $1.45 \times 10^{16} \text{ cm}^{-3}$ to avoid extrinsic effects such as the phonon bottleneck effect and multi-particle Auger recombination.⁵ By comparison of the TAS data before and after processing, we observed a decrease in the bleach signal intensity near the bandgap after processing, and the bleach band became broader and redshifted with respect to that before processing. Additionally, the photoinduced

absorption band covering a broad long-wavelength spectrum got a significant enhancement by laser-processing within 1 ns. These observations indicate the emergence of shallow trap states energy levels, accompanied by a decrease in the population of free carriers and an increase in the population of excitons after femtosecond laser processing. These changes are believed to be responsible for the photoluminescence enhancement observed within the microstructures.

Figure R19. The analysis of the PL mechanism of bulk crystal and microstructure of MAPbBr₃. (a) Theoretical calculated energy levels around the band gap for various defect structures for CsPbBr₃.³ The V_{Br} (V: vacancy) and P_{bi} (i: interval) defects introduce occupied deep levels inside the band gap (red lines). When O₂ is adsorbed onto the defect, the deep defect levels are removed. (b) FM images and the corresponding micro-PL spectra of processed MAPbBr₃ at different scanning speeds. (c) The relaxation dynamics of free carriers and excitons were investigated before and after laser processing, with probing wavelengths of 555 nm and 800 nm, respectively.

To further confirm this viewpoint, based on the reviewer's suggestion, we conducted intensity-dependent TAS tests on pristine MAPbBr₃ wafers. The results are shown in **Figure R20 & R21**. Firstly, we observed that while the bleach signal intensity increased with the increasing in excitation intensity, there was no significant peak shift (**Figure R21a**). In contrast, the TA spectra of the laser-processed crystals exhibited a clear redshift in the bleach signal (**Figure R18f**), indicating the presence of shallow trap states energy levels rather than a band-gap renormalization effect caused by the excited state carrier population near the bandgap. (*The extension of the bleaching signal at longer wavelength indicates the presence of shallow trap state. The unoccupied trap states permit weak optical transition from the ground states, resulting in optical absorption before photoexcitation. Under pump excitation the trap states have already been populated; thus,*

reduced ΔOD is expected.” Anita Ho-Baillie et al. *J. Phys. Chem. C* **2016**, *120*, 2542) Secondly, we noticed a positive correlation between the intensity of the photoinduced absorption band and that of the bleach signal, with both increasing or decreasing simultaneously (**Figure R21a**). Conversely, for the TA spectra of the laser-processed crystals, the bleach signal intensity decreased noticeably accompanied by the increasing of the absorption band (**Figure R18f**). This observation indicates a redistribution of the population of free carriers and excitons in the newly formed micro-/nano-structures after femtosecond laser processing, with a notable increase in the population of excitons. The analysis of these results is consistent with our previous hypothesis.

Figure R20. Pseudo colour TA plots on untreated MAPbBr₃ wafer at different excitation densities.

Figure R21. Effect of excitation density on transient absorption spectroscopy and

excited state carrier relaxation dynamics. (a) TA spectra at delayed 1.5 ps and **(b)** time-resolved relaxation dynamics at 555 nm for MAPbBr₃ crystal.

In conclusion, by utilizing complementary techniques including photoluminescence spectroscopy, transient absorption spectroscopy and X-ray photoelectron spectroscopy on MAPbBr₃ crystals before and after laser processing, compelling evidences have been provided to support the mechanism about the dramatic photoluminescence enhancement induced by laser processing of MAPbBr₃ crystals. The enhancement can be attributed to several key factors: (i) passivation of deep-level traps of metallic Pb atoms by oxygen exposure, (ii) the presence of shallow-level traps that act as beneficial recombination centers induced by femtosecond laser processing, and (iii) an increase in the population of excitons due to the size confinement effect of the microstructure. Collectively, these factors play a crucial role in the significant boost observed in the photoluminescence of the MAPbBr₃ crystal. The simplified flat-band energy level diagram model is depicted in **Figure R18g & h**, while a more detailed and intuitive explanation is presented in Figure 4o of the main text.

The above discussions are added to the main text and supporting information file, and are highlighted in red. Figure R18 is added into Figure 4 of the main text to replace the original Figure 4g-n. Figure R19c is added as Figure S9. Figure R20 and Figure R21a are added as Figure S12.

References

- 1 Lindblad R, Jena N K, Philippe B, et al. Electronic structure of CH₃NH₃PbX₃ perovskites: dependence on the halide moiety [J]. *The Journal of Physical Chemistry C*, **2015**, 119(4): 1818-1825.
- 2 Cho H, Jeong S H, Park M H, et al. Overcoming the electroluminescence efficiency limitations of perovskite light-emitting diodes [J]. *Science*, **2015**, 350(6265): 1222-1225.
- 3 Lu D, Zhang Y, Lai M, et al. Giant light-emission enhancement in lead halide perovskites by surface oxygen passivation [J]. *Nano letters*, **2018**, 18(11): 6967-6973.
- 4 Li D, Meng Y, Zhang Y, et al. Selective Surface Engineering of Perovskite Microwire Arrays [J]. *Advanced Functional Materials*, **2023**: 2302866.
- 5 Kaniyankandy S. Evidence of Auger heating in hot carrier cooling of CsPbBr₃ nanocrystals [J]. *Colloids and Surfaces A: Physicochemical and Engineering Aspects*, **2022**, 635: 128025.

REVIEWER COMMENTS

Reviewer #2 (Remarks to the Author):

The authors have considerably improved the discussion, with some better quality data, and addressed some of my comments. I particularly appreciate the detailed and well written response. However, a few concerns still persist even in this revision:

1. The assignment of the PA at 1.55 eV to exciton absorption is a bit weak. Firstly, as noted in PRL, 114, 116601, the exciton PA must be related to specific transitions between distinct excitonic states. For the lead system, this matches with the two excitons arising from two bands, which are separated by about 600-700 meV and accordingly the PA arises in the mid-IR range. While not many reports exist for the bromide counterparts, as noted by the authors JPCC 120, 2542 makes attempts to assign the rather small PA band to the excitons. Notably, that work does not focus entirely on this assignment, and it can only be taken as a possible suggestion. The authors should acknowledge this in their manuscript.
2. Assuming that the PA indeed corresponds to exciton, the symmetry considerations that determine the associated selection rules between excitonic states may not reflect that of free carriers. Polarization anisotropy directly reflects this and this manuscript needs more data at other probe wavelengths (and preferably taken at different pump wavelengths) to properly assess the anisotropy.
3. Looking at the maps in Figure 2f,g and h, it seem that the anisotropy dynamics manifest in the 100 picosecond timerange. Contrary to what the authors suggested, recombination dynamics can indeed be present in this situation, primarily through Auger processes. Can the anisotropy be related to many-body scattering, which may manifest differently in each of the crystal orientations?
4. The anisotropy data is very interesting, but with limited data, it lacks insightful discussion. Importantly there is no direct relevance of this intriguing observation in the femtosecond processing section. The latter is about defect physics and recombination dynamics (which happen in the sub-nanosecond timescales, Figure 4). While the proposed mechanism of PL enhancement is interesting, the evidence of shallow defect creation is weak. Simple intensity dependent PLQYs (and transient dynamics if possible) will substantially add to the discussion.

In my opinion, there are two independent yet very interesting observations in this manuscript, but lacks enough data to add depth to the discussion.

Response to the Comments and Suggestions of Reviewer 2

The reviewer's comments: *The authors have considerably improved the discussion, with some better quality data, and addressed some of my comments. I particularly appreciate the detailed and well written response. However, a few concerns still persist even in this revision.*

Authors reply: We greatly appreciate the reviewer for dedicating his\her time to review our manuscript and for providing positive feedback on our versions. We have carefully considered all of the comments and suggestions which are invaluable to improving the quality of our work. Below, we have outlined our responses to each point of the reviewer's concerns, and we are grateful for the opportunity to address them.

1. *The assignment of the PA at 1.55 eV to exciton absorption is a bit weak. Firstly, as noted in PRL, 114, 116601, the exciton PA must be related to specific transitions between distinct excitonic states. For the lead system, this matches with the two excitons arising from two bands, which are separated by about 600-700 meV and accordingly the PA arises in the mid-IR range. While not many reports exist for the bromide counterparts, as noted by the authors JPCC 120, 2542 makes attempts to assign the rather small PA band to the excitons. Notably, that work does not focus entirely on this assignment, and it can only be taken as a possible suggestion. The authors should acknowledge this in their manuscript.*

Authors reply: We express our gratitude to the reviewer for the thoughtful comments regarding the assignment of the photoinduced absorption (PA) signal at 1.55 eV to exciton absorption.

Firstly, about the PA band centering around 750 nm in MAPbBr₃, besides the report of [JPCC 2016, 120, 2542], wherein this near-infrared PA band was excited with a pump wavelength of 400 nm, (**Figure R1a**) the authors in [Adv. Energy Mater. 2016, 6, 1600551; Optica 2021, 8, 735] also recorded a similar PA band at this waveband using the same 400 nm pump; (**Figure R1b and c**) furthermore, when using an excitation wavelength of 540 nm, a more prominent PA band around 750 nm was observed in the literature [Adv. Optical Mater. 2018, 6, 1700975] (**Figure R1d**). Given that this excitation wavelength aligns closely with the energy band gap, supplementary experiments were conducted with a 515 nm excitation wavelength to conduct TAS measurement on a same MAPbBr₃ wafer. The results shown that compared to the case of using a pump wavelength of 400 nm, (**Figure R1e**) 515 nm excitation indeed generated a more prominent PA band at ~ 750 nm. (**Figure R1f**) Thus both our experimental results and the literature reports collectively confirm the emergence of this near-infrared PA band in MAPbBr₃, even under different pump wavelengths.

Secondly, about the attribution of the PA band, the authors of [JPCC 2016, 120, 2542] regarded it “*is most likely to be the absorption of photogenerated excitons*” based on the

analysis of the exciton binding energy, the rise time of the PA band and the comparable case in the MAPbI₃;^[PRL 2015,114, 116601; PCCP 2015, 17, 14674 in JPCC] the authors of [*Adv. Energy Mater.* **2016**, 6, 1600551] considered the PA band “possibly arises from the transitions of the photoexcited species to higher excited states or sub-bandgap trap state absorption in the crystal, which is significant in the very thick crystal (≈ 3.2 mm) while negligible in the thin film (≈ 100 nm)”; the authors of [*Optica* **2021**, 8, 735] regarded it “For single-photon pumping at 400 nm, we include only the exciton absorption spectrum...”; and the authors of [*Adv. Optical Mater.* **2018**, 6, 1700975] concluded that “While we cannot eliminate all other mechanisms, we believe that polaron formation is the most likely hypothesis for our TA observation in perovskite bulk single crystals.” The above reports demonstrate that there is indeed divergence on the assignment of the PA band, just as the reviewer has pointed out. Here given the low amplitude and longer lifetime of the PA band, and the “soft” lattice nature of MAPbBr₃ crystal, it is more likely that free carriers, excitons, and polarons coexist in the photogenerated states.^{JPCC & AEM & Optica & AOM}

Figure R1. The TAS and excitonic states in CH₃NH₃PbBr₃. (a) TA spectra of MAPbBr₃ film showing a weak PA band (PA₂) with 400 nm excitation.^{JPCC} (b) The pseudo colour TA profile of MAPbBr₃ crystal with 400 nm excitation.^{AEM} (c) The pseudo colour TA profile of MAPbBr₃ crystal with single-photon excitation (400 nm).^{Optica} (d) The pseudo colour TA profile of MAPbBr₃ crystal with 540 nm excitation.^{AOM} (e) Pseudo colour TA plot of MAPbBr₃ with 400 nm excitation (our work). (f) Pseudo colour TA plot of MAPbBr₃ with 515 nm excitation (our work). (g) Optical absorption spectrum of CH₃NH₃PbI₃ at 4.2 K.^{J. Phys. Soc. Jpn.} (h) Optical absorption spectrum of CH₃NH₃PbBr₃ at 5 K.^{Solid State Commun.}

When assuming the PA band around ~ 750 nm was attributed to exciton absorption, we agree with the reviewer that it's important to relate the exciton PA to specific transitions

between distinct excitonic states, as mentioned in [PRL 2015, 114, 116601]. **Figure R1g and h** represent the discrete excitonic lines in the optical absorption spectra of MAPbI₃ at 4.2K^[J. Phys. Soc. Jpn. 1994, 63, 3870] and MAPbBr₃ at 5 K^[Solid State Commun. 2003, 127, 619]. Based on the identified excitonic structures the authors indicated the PA band at ~ 600-700 meV originating from transitions between excitonic states E_I to E_{II} in MAPbI₃.^{PRL} For MAPbBr₃, the E_I and E_{III} exciton states, separated by an energy of ~1.642 eV, coincidentally corresponds to the PA band at ~750 nm (~1.650 eV) in the TAS. Hence, it is reasonable to speculate that the PA band is attributed to the inter-band exciton absorbing transition (E_I to E_{III}).

Whatever, at present all the assignments of the longer wavelength PA band are short of direct and solid evidence. In fact, even the exact band structures and the actual fundamental band-gap values of such hybrid perovskites are still under intense debate in the computational community.^[J. Phys. Chem. Lett. 2017, 8, 5507; J. Chem. Theory. Comput. 2016, 12, 3523] To quantitatively assign the features in the TA spectra to specific transitions through theoretical calculations is infeasible. Thus although it is considered that exciton absorption is a most likely hypothesis for the observed longer wavelength PA in perovskite bulk single crystals, we cannot eliminate the possibilities from other attributions such as polarons or other sub-bandgap trap state absorption. And the predictability of the specific transitions necessitates more detailed experimental and theoretical investigations in this field.

According to the reviewer's suggestion, we have acknowledged the uncertainty in assigning the PA band to excitons in the main text of the revised manuscript, page 6: "*a polarized probe pulse 1.55 eV arriving at various delay time is adopted to probe the anisotropic dynamical evolution of the photogenerated species. A photoinduced absorption band located around this wavelength in TAS (Figure S4), which emerged under different pump wavelengths (400 nm and 515 nm), is speculated to originate from inter-band transitions of excitons or localized excitons.²⁹⁻³¹ Given the low amplitude and longer lifetime of the photoinduced absorption band, and the "soft" lattice nature of MAPbBr₃ crystal, it is most likely that free carriers, excitons, and polarons coexist in the photogenerated states, thus we cannot eliminate the possibilities from other attributions to this band. Detailed assignment analysis see Supplementary Note 1.*"

The above discussions are added to **Supplementary Note 1** of the supporting information file, and **Figure R1e and f** are added as **Figure S4**.

2. Assuming that the PA indeed corresponds to exciton, the symmetry considerations that determine the associated selection rules between excitonic states may not reflect that of free carriers. Polarization anisotropy directly reflects this and this manuscript needs more data at other probe wavelengths (and preferably taken at different pump wavelengths) to properly assess the anisotropy.

Authors reply: Thanks for the reviewer's constructive suggestions. We agree with the reviewer's concern about the polarization anisotropy detected at the wavelength of 800 nm, which may predominantly reflect the selection rules between excitonic states rather than those of free carriers. Therefore, based on the transient absorption spectra of MAPbBr₃ crystals, we measured the dynamics at a wavelength of 575 nm that located on the bleaching band, to better capture the relaxation processes of free carriers, as shown in **Figure R2**.

Figure R2. Pseudo colour TA plot of MAPbBr₃ with 400 nm excitation.

The carrier dynamics probing at 575 nm on differently-oriented MAPbBr₃ wafers were conducted with the same test conditions as that in 800 nm detections (a 3.1 eV femtosecond pump laser and a pump fluence of 71 μJ cm⁻²). The bleaching dynamics at 575 nm (shown in the insets of **Figure R3**) can also be fitted well by two shorter lifetimes: τ_1 within ~ 20 ps and τ_2 within ~ 150 ps, and another longer lifetime, τ_3 , being fitted at the nanosecond scale. The overall time scale of the bleaching dynamics is slightly longer than that of the PA dynamics, but the time of peak occurrence is almost identical to the PA band, both around 1.5 ps, corresponding to the electron-electron scattering and electron-longitudinal optical (LO) phonon scattering.

Figure R3 shows the pseudo-color polarization-resolved transient absorption plots of the bleaching dynamics at 575 nm on the (100), (110) and (111) wafers. As can be seen for each stage of the carrier dynamics, including the extremely fast cooling stage represented by the peak value of ΔA , and the two decay stages represented by the lifetimes (τ_1 and τ_2),

they indeed demonstrated obvious angle-dependence upon variation of the probe polarizations. Compared to the anisotropy of dynamics probed at 800 nm, the anisotropic behaviors probed at 575 nm shows similar polarization-dependent symmetry, with a milder contrast among different orientations. Compared to free carriers, excitons should possess relative more localized nature in crystal, thus stronger interactions with the lattice and phonons during the relaxation process should be anticipated, which makes them more susceptible to the influence of the lattice structure. On the one hand, the additional subtle variations may be caused by inevitable light disturbances owing to the close proximity of signals at different polarization angles during measurements.

Figure R3. Anisotropic relaxation dynamics evolution of MAPbBr₃ probed at 575 nm.

Overall, the dynamics probing at the beaching band of 575 nm should reflect more information about free carriers of the excited state of MAPbBr₃. Although there are some differences in the relaxation lifetimes and polarization-dependent symmetry patterns, both the dynamics of free carriers and excitons exhibit obvious in-plane polarization dependence on (100) and (111) wafers while a weak dependence on the (110) wafer. Because the relaxation processes for both the two types of particles involve interactions with lattice vibrations (phonons) in the decay to lower energy levels or the ground state, they are influenced by the intrinsic factors such as the symmetry of electronic band structure, phonon spectra, and other lattice-related elements. Therefore, similar selection rules may take effect during the relaxation process regardless of the particle type.

According to the reviewer's suggestion, we have added the following text in the main text of the revised manuscript, page 8: "*We also measured the dynamics evolution on the three wafers at a wavelength of 575 nm that located on the bleaching band in TA spectra (Figure S4a), to reflect more information about the relaxation processes of free carriers in MAPbBr₃. The results show similar polarization-dependent symmetry, with a milder contrast among different orientations compared to that of the 800 nm detection, indicating the similar selection rules during the relaxation process regardless of the particle type. Detailed analysis see Supplementary Note 2.*"

The above discussions are added in the supporting information file as **Supplementary Note 2**, and **Figure R3** is added as **Figure S11**.

3. *Looking at the maps in Figure 2f,g and h, it seem that the anisotropy dynamics manifest in the 100 picosecond timerange. Contrary to what the authors suggested, recombination dynamics can indeed be present in this situation, primarily through Auger processes. Can the anisotropy be related to many-body scattering, which may manifest differently in each of the crystal orientations?*

Authors reply: We thank the reviewer for the comprehensive and in-depth comments.

It is indeed just as the reviewer has pointed out, the anisotropy dynamics of the photoexcited carriers manifest in the initial 100 picoseconds. Actually besides the pseudo-color maps of Figure 2f, g and h, from the contour plots of the extracted lifetimes (τ_1 and τ_2) in Figure 2l and o (for (100) wafer), Figure 2m and p (for (110) wafer), and Figure 2n and q (for (111) wafer), we could perceive that the first dynamic process (τ_1) occurred in less than 10 picoseconds and the longer process (τ_2) occurred in less than 100 picoseconds. In such a time range, our study mainly concerns the ultrafast processes of photoexcited hot carriers dynamics, including exciton\polaron formation, hot carrier cooling to the band-edge, and the subsequent regime involving the dynamics of "cold" carriers near the band-edge (including photoinduced lattice expansion, stress, strain, and coherent phonon effects) before their eventual trapping and recombination [*Chem. Rev.* **2023**, *123*, 8154]. Because these ultrafast processes are temporally overlapping, we agreed with the reviewer that recombination cannot be excluded from a 100-picosecond time range.

As to the specific recombination channel, it strongly relates to the excitation intensity, morphology, exciton binding energy, temperature, etc. Here in our carrier dynamic study, we used a low pump intensity (with an initial carrier density n_0 of $1.43 \times 10^{15} \text{ cm}^{-3}$). Because Auger recombination only dominates the carrier recombination processes at

higher carrier densities ($>10^{18} \text{ cm}^{-3}$), as it is a three-particle many-body process with a lower probability, [*Acc. Chem. Res.* **2016**, *49*, 166; *Nat. Commun.* **2017**, *8*, 14350; *Adv. Mater.* **2023**, *35*, 2301834] we regard Auger recombination may be negligible in our experimental cases. Moreover, Auger processes are generally more dominant in nanocrystals and quantum dots with confinement effects of carriers than in the bulk ones [*Angew. Chem. Int. Ed.* **2020**, *59*, 14292; *Nat. Phys.* **2006**, *2*, 557]. Thus in our high-quality bulk single crystalline MAPbBr₃ with long carrier diffusion lengths and moderate exciton binding energy, the first-order trap-mediated monomolecular recombination and the second-order bimolecular recombination would dominate the recombination process under such a low excitation intensity.

In the main text of the manuscript, page 8, we have pointed out that “*The low pump excitation (with an initial carrier density n_0 of $1.43 \times 10^{15} \text{ cm}^{-3}$) makes the carrier relaxation dynamics be less influenced by extrinsic effects such as the phonon bottleneck effect and the multi-particle Auger-recombination.*” We tried to probe the intrinsic carrier-phonon interactions to unravel the relations between the carrier dynamics and lattice orientations based on anisotropic crystals.

4. *The anisotropy data is very interesting, but with limited data, it lacks insightful discussion. Importantly there is no direct relevance of this intriguing observation in the femtosecond processing section. The latter is about defect physics and recombination dynamics (which happen in the sub-nanosecond timescales, Figure 4). While the proposed mechanism of PL enhancement is interesting, the evidence of shallow defect creation is weak. Simple intensity dependent PLQYs (and transient dynamics if possible) will substantially add to the discussion. In my opinion, there are two independent yet very interesting observations in this manuscript, but lacks enough data to add depth to the discussion.*

Authors reply: We appreciate gratefully the reviewer for recognition of our work as “*very interesting*”, including both the “*intriguing*” anisotropy data and the “*very interesting observations*” of the processing induced PL enhancement. And also, the reviewer proposed constructive suggestions to make a more insightful discussion. Following these suggestions, we have supplemented intensity dependent PLQY and TAS measurements to add depth to the discussion.

In **Figure R4a**, we show the experimental steady-state PLQY data as a function of excitation density of MAPbBr₃ crystals before and after processing, respectively. We note that the pristine MAPbBr₃ (unprocessed bulk crystal) exhibits a very low PLQY of $<0.1\%$

under the measurement conditions with excitation densities lower than 10^{16} cm^{-3} . (**Figure R4d**) While the processed crystal shows a much higher quantum yield in the order of ten percent at similar densities (and under ambient atmosphere). (**Figure R4e**) We find that the PLQY of the pristine MAPbBr₃ crystal shows an upward change at high excitation fluences. This trend is consistent with the model described in ref [*Phys. Rev. Appl.* **2014**, 2, 034007] that the charge-trapping pathways limit the radiative recombination at low excitation fluences, and the radiation would be significantly enhanced when the charge trap states are filled by photogenerated carriers at higher excitation fluences. Additionally, we have confirmed the presence of metallic Pb(0) defect in the pristine bulk MAPbBr₃ and its transformation into oxygen passivated-Pb(II) after laser processing by X-ray photoelectron spectroscopy (XPS). Because the excess Pb(0) atoms have been proved to act as deep defect levels that cause nonradiative decay to degrade the photoluminescence, it is suggested that its transformation into the oxygen passivated-Pb(II) corresponds to a transition from nonradiative deep-level traps to radiative shallow traps after laser processing. According to the model described in ref [*J. Am. Chem. Soc.* **2016**, 138, 13604], the participation of shallow defects in the radiative process leads to an increased PLQY compared to the case of deep trap states. Our PLQY measurements show a pronounced increase from the pristine crystal to the laser-processed one, indicating the formation of shallow-trap states by induction of femtosecond laser and/or passivation of the deep-level traps by ambient oxygen. Due to the limit of the power of excitation light source, the excitation intensity-dependence of the measured PLQYs for both the pristine and processed MAPbBr₃ remain pretty insensitive at the lower excitation intensity range ($<10^{16} \text{ cm}^{-3}$). However, a slight enhancement of the PLQY was still observed on the pristine crystal at higher excitation intensity. This slight increase of PLQY may be attributed to the increased filling of trap states and, possibly, to an increasing excitonic fraction of photogenerated species at higher excitation intensity. [*Phys. Rev. Appl.* **2014**, 2, 034007] It's reported the substantial increase of PLQY for the case of shallow traps may occur at even higher excitation intensity due to the high depopulation rate R_{dep} of the trapped carriers compared to that of the deep traps. [*J. Am. Chem. Soc.* **2016**, 138, 13604] It is noted that due to the fact that the measured PLQY of the pristine crystal is quite low, even within the degree of measurement error, the conclusion is indeed lack of solid evidences.

Figure R4. The intensity-dependent PLQYs and TAS studies. (a) PLQY data of MAPbBr₃ crystals before and after processing as a function of charge density. (b) TA spectra at different excitation densities at 1.5 ps of unprocessed MAPbBr₃ crystals. (c) TA spectra at different excitation densities at 1.5 ps of processed MAPbBr₃ crystals. (d) PLQY at excitation density of $6.03 \times 10^{15} \text{ cm}^{-3}$ for unprocessed MAPbBr₃ crystals. (e) PLQY at excitation density of $6.12 \times 10^{14} \text{ cm}^{-3}$ for processed MAPbBr₃ crystals.

The involvement of different types of defects is also supported by the intensity-dependent TAS measurements. **Figure R4b** illustrates that in the pristine crystals, the intensities of both the photoinduced bleach (PB) bands and the photoinduced absorption (PA) bands (exciton absorption band) exhibit an upward trend as the excitation density rises; on the other hand, for the laser processed MAPbBr₃ crystal, the PA bands intensity remains nearly constant as rising of the excitation density. (**Figure R4c**) Furthermore, considering the phenomenon we observed in the measurements that the pristine crystals only showed perceptible green emissions under very strong laser excitation, it indicates that deep-level defects dominate the recombination in the pristine bulk crystals. These deep-level defects need more photogenerated carriers to fill up, thus the signals of the exciton absorption become prominent under stronger excitation. While in the laser processed crystal, the exciton fraction from the photogenerated free carriers is high even under low excitation density because there is no such a pre-process filling. And because of the competition from radiative recombination via the shallow traps, the population of the photogenerated excitons may keep in a relatively steady level with increasing of the excitation fluences.

This observation is further corroborated by the redshift in the bleaching band compared to the pristine crystal. The authors of [JPCC 2016, 120, 2542] regarded it “*The extension of the bleaching signal at longer wavelength indicates the presence of shallow trap state.*” Overall, the creation of shallow-trap states induced by femtosecond laser and/or transformed from passivation of the deep traps by ambient oxygen are considered to be one of the key factors responsible for the observed enhancement in PL.

Given the main topic of this work—exploring the interaction of ultrafast laser with single crystal perovskites, on the one hand, anisotropic ultrafast carrier dynamics on differently oriented crystal planes were firstly revealed; and on the other hand, the mechanism of the PL enhancement induced by laser processing is uncovered by deciphering the ultrafast carrier dynamics. Between the two parts of discoveries, the common research technique, time-resolved pump-probe, and the common anisotropic behavior in both ultrafast dynamics and steady-state photoluminescence link them together. The supplementary experiments and discussions help to strengthen the relevance between the ultrafast dynamics and the mechanism of PL enhancement induced by the femtosecond laser processing. We have been committed to addressing the concerns of the reviewer by incorporating the suggested experiments and discussions to add depth to our findings.

In the main text of the manuscript, page 13, we have pointed out that “*The presence of shallow trap states energy levels and the redistribution of the population of free carriers and excitons are further verified by intensity-dependent PLQYs and TA spectra (Supplementary Note 5).*”

The above discussions are added in the supporting information file as **Supplementary Note 5**, and **Figure R4** is added as **Figure S13 and 15**.

REVIEWERS' COMMENTS

Reviewer #2 (Remarks to the Author):

Firstly, I appreciate the work done by the authors in the revision of the manuscript. While they have addressed most of my concerns, I still believe that there are two independent yet interesting observations reported. Having said that, they results are worthy to be published and the manuscript can be considered for publication in the current form.